**communications** engineering

# Polarization sensing of network health and seismic activity over a live terrestrial fiber-optic cable
Charles J. Carver ✉ & Xia Zhou

Wide-scale sensing of natural and human-made events is critical for protecting against environmental disasters and reducing the monetary losses associated with telecommunication service downtime. However, achieving dense sensing coverage is difficult, given the high deployment overhead of modern sensor networks. Here we offer an in-depth exploration of state-of-polarization sensing over fiber-optic networks using unmodified optical transceivers to establish a strong correlation with ground truth distributed acoustic sensing. To validate our sensing methodology, we collect 85 days of polarization and distributed acoustic sensing measurements along two colocated, 50 km fiber-optic cables in Southern California. We then examine how polarization sensing can improve network reliability by accurately modeling overall network health and preemptively detecting traffic loss. Finally, we explore the feasibility of wide-scale seismic monitoring with polarization sensing, showcasing the polarization perturbations following low-intensity earthquakes and the potential to more than double seismic monitoring coverage in Southern California alone.

As of 2021, optical fiber networks reached over 43% of United States households, demonstrating a tremendous 12% yearly growth and projected to continue rising[1]. This fast adoption rate is unsurprising given the communication benefits of optical fiber over traditional copper networks, namely faster speeds and greater bandwidth[2,3]. In addition to these communication benefits, optical fiber networks hold a powerful latent ability: the capacity for wide-scale sensing of both human-made and environmental events, critical for reducing economic loss and protecting against ecological disasters[4,5].

Various forms of optical fiber sensing have been considered since the 1960s[6,7], all designed to sense manifold optical properties with various hardware. Contemporary methodologies include optical interferometry[8–11], microwave frequency interferometry[12], inelastic scattering based techniques (e.g., Brillouin optical time-domain analysis[13] and Raman optical time-domain reflectometry[14]), coherent Rayleigh backscatter-based distributed acoustic sensing (DAS)[4,15], and state-of-polarization (SOP) sensing[16–19]. To start, optical interferometry utilizes a highly stable laser source to measure the phase of light traversing through fiber, sensing disturbances caused by environmental events such as seismic waves. Microwave frequency interferometry also senses phase, specifically from a microwave-frequency modulated optical carrier, which relaxes the requirement of an ultra-stable laser source but necessitates dedicated hardware for optical mixing and modulation. Brillouin optical time-domain analysis (BOTDA) leverages stimulated Brillouin light scattering to linearly measure strain and

temperature impacting the fiber. In contrast, Raman optical time-domain reflectometry (ROTDR) is predominately used for temperature measurements. Coherent Rayleigh backscatter-based DAS senses both phase and amplitude but typically requires specialized hardware and dedicated fiber. Finally, SOP sensing measures the light's Stokes parameters—a four-dimensional vector describing the light's entire state-of-polarization[20]—over time by leveraging the internal digital-signal-processing-based polarimeters of telecom-grade coherent optical transceivers, thus requiring no infrastructure modifications or dedicated hardware.

Ongoing commercial sensing efforts have overwhelmingly tended towards DAS, given its high sensitivity and accurate spatial localization over kilometer distances. DAS uses optoelectronic interrogator devices that inject short pulses of light into fiber-optic cables and then measure optical distortions in the backscattered light, thereby deriving strain-rate signals proportional to the amount of physical stress affecting the fiber[14,21–23]. Because light travels in an optical fiber at an almost constant velocity (≈0.2 m/ns), simple time-to-distance calculations allow the strain signals to be localized with meter accuracies along the fiber. This capability permits the conversion of several kilometers of fiber-optic cables into thousands of seismo-acoustic sensors, making it a powerful tool for monitoring various mechanical responses. On terrestrial fiber routes, DAS has become the gold standard for pipeline surveillance[24], perimeter protection[25], traffic monitoring[26], environmental sensing[27–30] and, most recently, network health monitoring[16]. In underwater environments, DAS has also been established

Columbia University, 500 W 120th St, New York, NY, USA. ✉e-mail: cjc@cs.columbia.edu

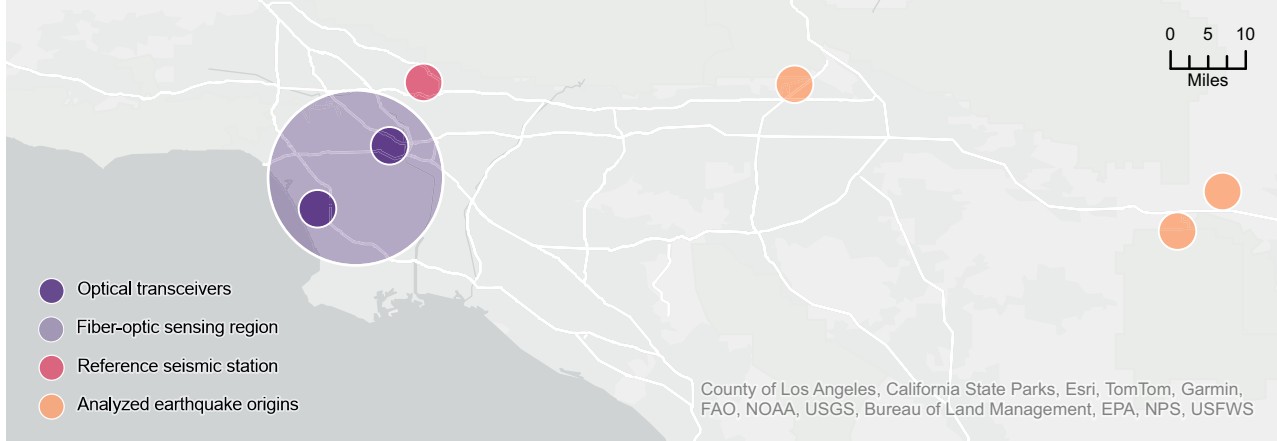

**Fig. 1 | Geographical placement of experimental setup in Southern California.** Our sensing setup comprises a 50 km fiber-optic cable connecting two optical transceivers in Inglewood, CA, and Downtown Los Angeles. We investigate the fidelity of state-of-polarization (SOP) sensing by comparing polarization measurements to ground-truth distributed acoustic sensing (DAS) samples along a parallel fiber running from Inglewood (DAS channel 0) to Downtown Los Angeles (DAS channel 4700). Additionally, we analyze the SOP fluctuations that follow the earthquake epicenters marked in orange, leveraging the nearby CI PASC[68] seismic station for waveform comparison.

as a reliable method for carrying out coastal geotechnical surveys and for monitoring environmental processes, such as earthquakes, ocean currents, and volcanic events[16,28,31–35].

Notwithstanding its popularity, DAS has considerable monetary and technical overheads that preclude its broader adoption. The expenditures are mainly driven by the prohibitive manufacturing costs of high-power laser interrogators[36] and the fact that DAS typically requires a dedicated dark fiber (i.e., optical fiber used solely for sensing without any communication channels[27,37,38]) to operate, thereby limiting the overall data-carrying capacity of the network. Recent work has shown the potential for both constant-amplitude DAS and L-band DAS over live fiber, but these systems still require costly DAS hardware or custom FPGA-based systems for signal injection[39,40]. Additionally, DAS is incompatible with inline optical amplifiers common to fiber routes, as the amplifier's optical isolator blocks the backscattered DAS signal. Although the optical amplifiers could be intentionally removed along the dedicated dark fiber, the lack of amplification would lead to rampant signal attenuation, necessitating multiple DAS devices to cover a sufficiently large sensing area (for example, inline amplifiers occur approximately every 80 km).

In response to these challenges, SOP sensing has emerged as a viable DAS alternative. Unlike DAS and interferometric systems, SOP sensing analyzes either the integrated[16–18] or localized[19,41–44] polarization changes of the modulated light traversing through traffic-carrying optical fibers. Importantly, since modern optical transceivers intrinsically require SOP measurements to demodulate the optical data signals[45], the SOP Stokes parameters are available without requiring dedicated dark fiber or specialized hardware. These factors make SOP monitoring the most scalable sensing method to date[16,46], and perhaps the most efficient route for enabling a wide-scale fiber-optic sensing network. Despite the potential of SOP sensing, prior work has yet to evaluate its performance on terrestrial optical fiber networks using already deployed telecommunication equipment. Instead, SOP sensing has been primarily limited to detecting sub-sea environmental events (e.g., earthquakes and ocean waves), given the low noise level and long spans of underwater optical fiber[16,47–52]. Recent work[53] attempted to bring SOP sensing to terrestrial fiber networks, showing an initial qualitative correlation between DAS and SOP signals in the presence of fiber stress. This work, however, required dedicated hardware for sensing and only recorded data during a limited collection period. An extension of this work[49] demonstrated the ability to sense a network-impacting event over live fiber but again required dedicated hardware and lacked a DAS ground-truth comparison.

This work explores terrestrial SOP sensing over live optical fiber with unmodified coherent transceivers. First, we characterize terrestrial perturbations with lab-based measurements of typical fiber-optic stressors. Second, we collect 85 days of continuous SOP measurements and DAS ground-truth data over two colocated 50 km fiber spans in South California (Fig. 1), showing a strong cross-correlation between the two signals. Third, we demonstrate that terrestrial SOP sensing can strengthen network robustness by (1) accurately modeling overall network health and (2) preempting traffic loss with a 160 s decision boundary assuming real-time detection. Fourth, we examine leveraging SOP sensing for passive seismic monitoring, illustrating the SOP perturbations generated by low-intensity earthquakes. Finally, we discuss the future of terrestrial SOP sensing and identify the critical next steps to fully realize its potential.

## Methods
### SOP primer
Light, being an electromagnetic wave, is composed of an orthogonal electric field, **E**, and magnetic field, **H**, that move along in the direction of light propagation. The polarization of light is then defined as the direction of its three-dimensional electric field vector. Assuming light propagates along the z-axis, linear polarization is caused by in-phase oscillations in the x–y plane. If a phase shift is introduced between the horizontal and vertical components, left- and right-handed elliptical polarization (including left- and right-handed circular polarization) can be created. A mathematically useful representation of this ellipse is using Stokes parameters, or the equivalent Stokes vector[20]:

$$\mathbf{S'} = \begin{pmatrix} S'_0 \\ S'_1 \\ S'_2 \\ S'_3 \end{pmatrix} = \begin{pmatrix} I \\ Ip\cos(2\psi)\cos(2\chi) \\ Ip\sin(2\psi)\cos(2\chi) \\ Ip\sin(2\chi) \end{pmatrix} \quad (1)$$

This formulation considers the total light intensity ($S'_0$), the shape of the polarization ellipse ($S'_1$, $S'_2$, and $S'_3$), and the degree of polarization $P = \sqrt{{S'_1}^2 + {S'_2}^2 + {S'_3}^2}/S'_0$ such that $0 \le P \le 1$. For light traveling through optical fiber, which is generated by a laser source and is therefore inherently linearly polarized, the degree of polarization is always $\approx 1$. Since the total light intensity ($S'_0$) is not crucial for sensing SOP perturbations, the Stokes

parameters can be normalized by dividing each component by $S_0'$, i.e.:

$$\mathbf{S} = \frac{1}{S_0'}\mathbf{S}' = \begin{pmatrix} S_1 \\ S_2 \\ S_3 \end{pmatrix} \qquad (2)$$

These normalized Stokes parameters can be visualized by plotting on the three-dimensional, unity-radius Poincaré sphere[20]. Given this representation, all completely polarized states of the light are represented as points on the sphere's radius, while unpolarized states lie within the sphere.

To help analyze SOP perturbations, we typically remove environmental/device drift by rotating the Stokes parameters to north[8] on the Poincaré sphere using the Padé approximation for matrix exponential[54]. The resulting rotated Stokes parameters are then represented as:

$$\mathbf{R} = \begin{pmatrix} R_1 \\ R_2 \\ R_3 \end{pmatrix} \qquad (3)$$

### Short-term-long-term averaging of SOP and DAS signals

To directly compare SOP and DAS signals, we employ the recursive short-term-average-long-term-average (STA/LTA) algorithm commonly used in seismic and acoustic emission analyses to detect anomalous events in continuous time series[28]. This algorithm calculates the average values of the absolute amplitude of a given signal in two consecutive moving time windows of different lengths and takes their ratio. The short-time window (STA) is designed to be sensitive to the anomalous nature of the instantaneous event, whereas the long-time window (LTA) gives information about the background noise of the system. When the ratio of these two values exceeds a certain empirical threshold, the algorithm classifies the timestamp of the STA window as the time of occurrence of an abnormal event. For a time-series vector $\mathbf{V}$ of $T$ measurements, an offline detection algorithm can be written as:

```
Data: V where V_i ∈ {0 . . . T}
Result: A where A_i ∈ {0 . . . T − M}
for t ← 0 to T − M do
    STA ← (1/N) Σ_{i=t}^{t+N} V_i
    LTA ← (1/M) Σ_{j=t}^{t+M} V_j
    if STA/LTA ≥ ε then
    |   A_t = 1
    else
    |   A_t = 0
    end
end
```

where $\mathbf{A}$ is a time-series vector of anomalous events, $N$ is the number of samples to average over in the short-time window, and $M$ is the number of samples to average over in the long-time window, and $\epsilon$ is the empirically chosen threshold.

In the context of SOP, after the Stokes parameters are rotated, $R_1$ and $R_2$ are centered around zero, and $R_3$ has a mean value around one. When performing STA/LTA measurements of these parameters, we subtract $R_3$ from one (i.e., $1 - R_3$) to ensure STA/LTA is equally sensitive to all parameters. In the context of DAS, we apply this technique to each DAS channel using an STA and LTA value of 0.1 s and 10 s, respectively. We then reduce the dimensionality of the output to a single time-series signal by taking the average STA/LTA value across all channels for each timestamp. The resulting signal thus provides us with an average metric of the abnormality of the DAS data as a function of time.

### Obtaining optical measurements

Crucial to sensing SOP perturbations, many modern optical transceivers directly measure the Stokes parameters with a polarimeter to compensate for polarization-dependent losses in fiber and, depending on the system, aid in demodulation when polarization-division multiplexing is implemented[45]. As a result, a complete representation of the light's SOP is freely available to already deployed optical transceivers. The Stokes parameters are exposed to our telemetry backend at an average 17 Hz sampling rate. We utilize an Optasense QuantX interrogator running the latest production build for DAS measurements.

## Results

### Lab validation of terrestrial SOP sensing

Prior work has extensively outlined the physics governing stress-induced polarization perturbations in sub-sea optical fiber[47], noting that hydrostatic pressure changes in the water column are the dominant factor in sub-sea environments, whereas bending and twisting are the dominant factors on land[47,53]. These terrestrial stressors are typical for physical cable maintenance activities and pose the highest risk to network uptime[55]. We experimentally validate the SOP sensing capabilities of detecting these stressors in a laboratory environment, utilizing a 15 km fiber spool connected to two optical transceivers. The optical transceivers expose the SOP measurements as a three-dimensional normalized Stokes vector $\mathbf{S} = \{S_1, S_2, S_3\}$, described at length in Methods. Within the laboratory environment, we test the SOP response to common fiber maintenance activities[56,57], namely: (1) bending the fiber to a tight radius, (2) bending the fiber to a wide radius, (3) moving the fiber spool, (4) pulling a section of fiber from within the spool, and (5) pulling the fiber's transceiver connector. In Fig. 2, we plot the three normalized Stokes parameters and corresponding spectrograms for each activity.

We observe a range of frequency and amplitude responses to the various stressors, with the highest frequency perturbations occurring when the fiber is bent or moved. Across all stressors, we observe the strongest PSD when the fiber is bent or pulled from within the spool and minor perturbations when the transceiver connector is pulled. This general observation agrees with the experimental results of refs. 49,53 and the analytical results of ref. 47, indicating that twists and bends in optical fiber are the most substantial source of SOP perturbations. Notably, the degree of polarization remains $\approx 1$ despite perturbations of the individual Stokes components.

Scrutinizing these results, however, we see a variable response to the same stressor event at different times. Unlike the stressors performed in ref. 41, which were constrained to well-defined motions, our stressors were not tightly regulated and followed typical motion patterns associated with fiber maintenance activities. To quantify the characterization ability from these measurements, we cross-correlate each stressor's waveform (per rotated Stokes parameter) with all seven waveforms to maximally align each stressor's temporal signature. Second, we compute the Pearson correlation coefficient for each combination of stressor events to construct a confusion matrix of correlations for each Stokes parameter. Third, we average the correlation coefficients for each combination across all Stokes parameters. This analysis shows the highest mean correlation ($r = 0.71$) between the tight and second wide bends and the lowest ($r = 0.18$) between the connector pull and the second fiber spool movement. Although this appears to indicate motions of different types (e.g., bending vs. pulling) can be reliably distinguished, stressors of the same kind often show relatively low correlations ($r = 0.50$ for the two different wide bends, $r = 0.21$ for two fiber spool movements). In contrast, different stressors show comparably high correlations ($r = 0.65$ for pulling the connector and the wide bend). This inability to reliably characterize based on temporal signatures is reasonable given the nonlinear response of the Stokes parameters to strain and the known fact that their evolution depends on the stochastic nature of the fiber's birefringence; classification is comparably more straightforward with linear strain-measuring techniques such as DAS, Brillouin, or microwave-based sensing. Precise characterization aside, we still observe an opportunity to identify atypical SOP activity, which we expand upon in the following sections in the context of a live, traffic-carrying fiber-optic cable in Southern California.

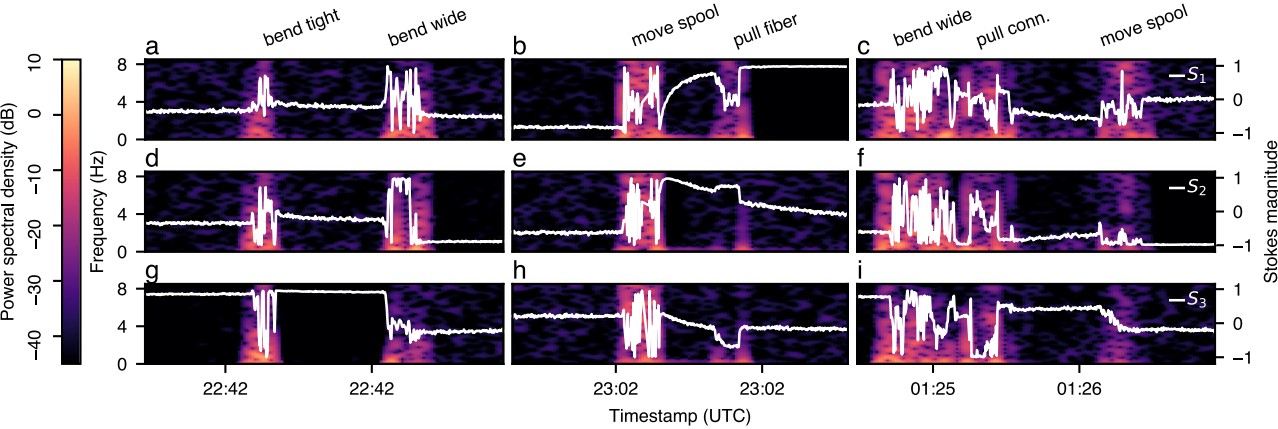

**Fig. 2 | State-of-polarization perturbations and corresponding spectrograms for common terrestrial stressors.** Various stressors (labels above **a–c**) are applied to a 15 km laboratory fiber-optic cable, causing the Stokes components to fluctuate. **a–c** show $S_1$ perturbations. **d–f** show $S_2$ perturbations. **g–i** show $S_3$ perturbations. Corresponding spectrograms are shown behind each Stokes parameter.

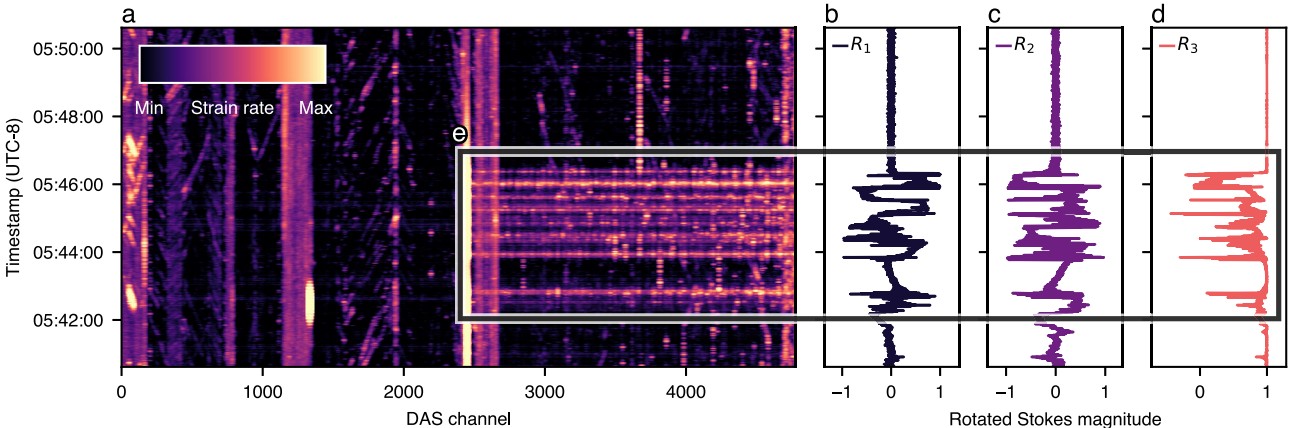

**Fig. 3 | Visual correlation between distributed acoustic sensing waterfall and rotated Stokes components.** The DAS waterfall (**a**) and Stokes components (**b, c, d**) capture the strain acting upon the experimental fiber on September 26th, 2022. The distributed acoustic sensing (DAS) channel represents the spatial location along the fiber. **e** highlights the strain event captured by both sensing methodologies, lasting roughly 6 min.

## Continuous SOP monitoring over a live optical fiber

Having shown the feasibility of sensing physical stressors with SOP perturbations, we next enable SOP sensing on two field-deployed optical transceivers in Southern California. The approximate length of the fiber route is 50 km, beginning in Inglewood, CA, and ending in downtown Los Angeles. This route was chosen since it has a DAS interrogator operating on a parallel dark fiber, thus enabling us to record continuous ground truth measurements for validating SOP sensing in a real-life environment. The DAS system is configured to record strain rate with a 10 m spatial resolution and gauge length (i.e., the distance between two points where phase measurements are made) of 20 m. SOP and DAS measurements were recorded from August 25th, 2022, to November 18th, 2022, for a total of 85 days.

Figure 3 compares coeval 10-min segments of DAS and SOP. The DAS measurements are displayed in their typical waterfall representation, where the $x$-axis corresponds to the channel number (i.e., spatial location along the fiber), the $y$-axis is the UTC timestamp, and the color represents strain rate. Regarding SOP, we rotate the Stokes vector toward north on the Poincaré sphere over a 10 s moving average time window (see the "Methods" section), similar to the work of ref. 46. This operation is designed to remove long-term polarization drifts caused by environment instability (e.g., temperature drifts[58]) and to center the two independent Stokes parameters, $R_1$ and $R_2$, around zero, resulting in an $R_3$ mean value of one. These rotated Stokes parameters are shown in the three panels on the right, with the $x$-axis showing the

magnitude of each Stokes component and the $y$-axis showing UTC timestamps.

A cursory inspection of Fig. 3 shows a remarkable agreement in time between the sequence of anomalous signals sensed by both technologies at 05:43. The strain event occurs around channel 2500, with each horizontal line indicating abnormal strain corresponding to an SOP perturbation. Aerial sections of the fiber route are shown as high-intensity vertical lines across the DAS waterfall. The impact of environmental stressors on aerial fiber sections is detailed in later sections. We quantify this empirical relationship between SOP and DAS by performing cross-correlations between the three SOP Stokes components and an average representation of DAS strain rate along the entirety of the fiber. To generate the averaged signal, we apply the short-term-average-long-term-average (STA/LTA) algorithm[28] to each DAS channel and take the mean across all channels (see the "Methods" section). On a high level, the STA/LTA algorithm normalizes the impact of instantaneous stressor events (captured by the short-time window, STA) relative to the longer-term background intensity (captured by the long-time window, LTA) to extract vital events from the nonlinear SOP response. For consistency, we apply the same STA/LTA algorithm to each Stokes component (as described in the "Methods" section) and resample both SOP and DAS STA/LTA measurements to a constant rate of 17 Hz, i.e., the average sampling rate of SOP and 34% of the average sampling rate of DAS. Finally, to ensure we only correlate sufficiently strong SOP/DAS signals and not merely noise, we only consider the signals within each 10-min interval

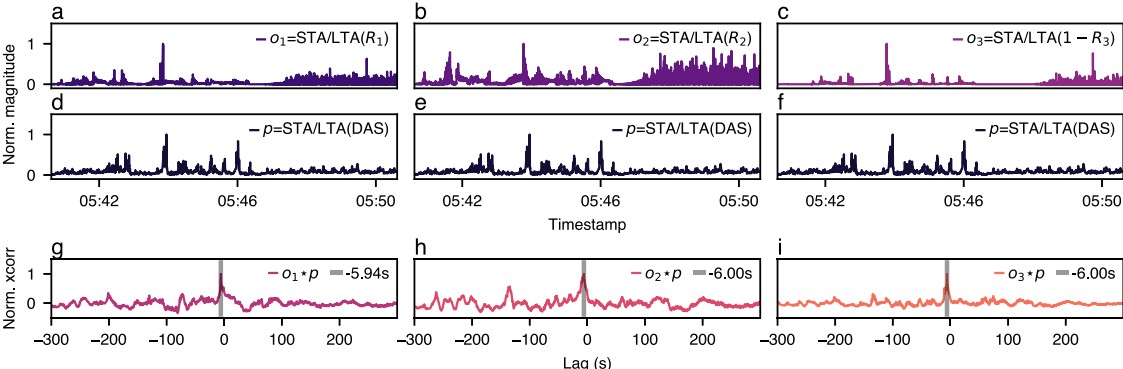

**Fig. 4 | Cross-correlations between short-term-long-term-average Stokes events and distributed acoustic sensing events detected on September 26th, 2022.** **a–c** show the time-series short-term-long-term-average (STA/LTA) Stokes events ($o_{1...3}$). **d–f** show the corresponding STA/LTA distributed acoustic sensing (DAS) events ($p$). **g–i** show the cross-correlations between SOP and DAS events ($o_{1...3} \star p$).

Notably, we leverage the STA/LTA algorithm to transform the two-dimensional DAS data into a one-dimensional signal. We apply the same transformation to the Stokes parameters for consistency and to account for the nonlinear polarization response to fiber strain.

surrounding high-strain activity (i.e., a STA/LTA value ≥3). A visual workflow of this correlation analysis can be seen in Fig. 4.

Across the $n = 604$ detected DAS events, the median peak cross-correlation lags are $-5.5$ s for $R_1$ (Q1 $= -4.7$ s, Q3 $= -7.1$ s), $-5.4$ s for $R_2$ (Q1 $= -4.8$ s, Q3 $= -6.1$ s), and $-5.6$ s for $1 - R_3$ (Q1 $= -4.3$ s, Q3 $= -6.6$ s), where Q1 is the 25th percentile (1st quartile) and Q3 is the 75th percentile (3rd quartile). This distribution includes all events over the months-long collection period, including those with lags ≥ the maximum observed Q3 value ($n = 78$). See Supplementary Fig. 1 for a visual representation of these statistics. Notably, since the SOP signals are used as the reference, these negative lags indicate that SOP measurements preceded DAS measurements by ≈5 s. The source of this discrepancy is the DAS interrogator and transponder, which need to be periodically synced with network time to prevent clock drift, a common issue in data network sensing. Specifically, when the DAS interrogator begins an acquisition period, its measurement timestamp will slowly drift from network time proportional to the acquisition length. The median peak lag (averaged across all SOP parameters) at the beginning of our data collection period was $-5.35$ s and $-6.02$ s at the end. This phenomenon is not encountered with SOP sensing since timestamps are measured directly by the network-synced transceiver.

Given the demonstrated accuracy of SOP sensing compared to the gold-standard DAS, it is natural to question which sensing methodology is most appropriate for a given situation. On the one hand, DAS has powerful temporal and spatial sensing capabilities, detecting a wide variety of stressor events over space and time. This comes at the cost of expensive equipment, high deployment overhead, and lower network bandwidths. On the other hand, SOP sensing provides a much simpler—yet still high fidelity—integrated picture of fiber stress affecting the entire optical fiber route. Although this integrated view discards spatial information and makes it challenging to physically localize stressors along the fiber, recent work has attempted to bring localization to SOP[19,41–44]. Regardless, SOP sensing requires no additional equipment, has zero deployment overhead, and has a negligible impact on network bandwidth. In the following sections, we will demonstrate two applications that SOP sensing is primed to support: strengthening network robustness and wide-scale seismic monitoring.

**Improving network reliability through SOP sensing**
In this section, we explore the first practical application of SOP sensing: strengthening network robustness. Within the telecom community, it is widely known that human maintenance of terrestrial optical fiber is a leading cause of network-impacting events, resulting in downtime for telecommunication customers and monetary losses for providers[59]. These activities typically include splicing closure tech operations, fiber over-pulls of existing infrastructure, and tech procedures at locations with active

networking activity[56,57]. These events often result in network flaps—i.e., momentary network outages caused by scheduled maintenance or sufficiently strong stressor events that bring the network offline[60,61].

Frequently, these maintenance activities impact fiber strands in a cable or conduit in addition to the strands undergoing scheduled work. In this case, the network operator is not warned before the planned maintenance, meaning a flap may have a greater impact on traffic and services carried over it. In these cases, the benefit of SOP detecting this activity and the potential of a flap or fiber cut in the next several minutes or hours is beneficial in at least two ways: (1) the network operator can divert high-priority traffic to a route that is not experiencing this activity; (2) the fiber network provider can be contacted to adjust the maintenance activity to stop it from impacting additional fiber strands.

While DAS has recently been used to detect these events and provide actionable insights into preventing future downtime[16], its monetary and deployment overhead has prevented it from becoming a large-scale solution. This section demonstrates that SOP sensing is a viable, passive alternative to DAS that can seamlessly be deployed to entire networks. Integrated SOP sensing is valuable in this context as it does not inherently require linearity or spatial resolution and offers increased ease of access over alternative methods.

**Preemptive detection of network flaps**. We begin by demonstrating that SOP sensing can detect network flaps. Figure 5 shows the SOP measurements when a network flap occurs along the fiber on November 15th, 2022, at 7:34 UTC (midnight local time, UTC-8). Throughout the majority of the 30-h window, the Stokes parameters are relatively stable. However, 20 min before the flap event, the Stokes parameters begin rapidly fluctuating with transient periods of stability. Given these perturbations' temporal, frequency, and time-of-day characteristics, the flap was most likely caused by maintenance on a parallel fiber-optic cable. Importantly, not only is SOP sensing able to profile the stress impacting the fiber (i.e., having a similar frequency response as wide-angle fiber bends and connector pulling as shown in Fig. 2), but it also detects the perturbations preceding the flap event. This realization paves the way for improved traffic routing algorithms to preemptively reroute packets when specific SOP perturbations are observed, thereby improving overall network stability and uptime for clients and providers. As an initial exploration into preemptive rerouting, we design a linear classifier to differentiate between perturbation windows with and without flaps. As illustrated by Fig. 5, we can identify two characteristics that distinguish between these two categories: the duration of the encompassing perturbation window and the strength of perturbations within the window. First, we automatically detect perturbation windows by taking the first derivative of one SOP channel, e.g., $S_1$, which extracts periods of rapid

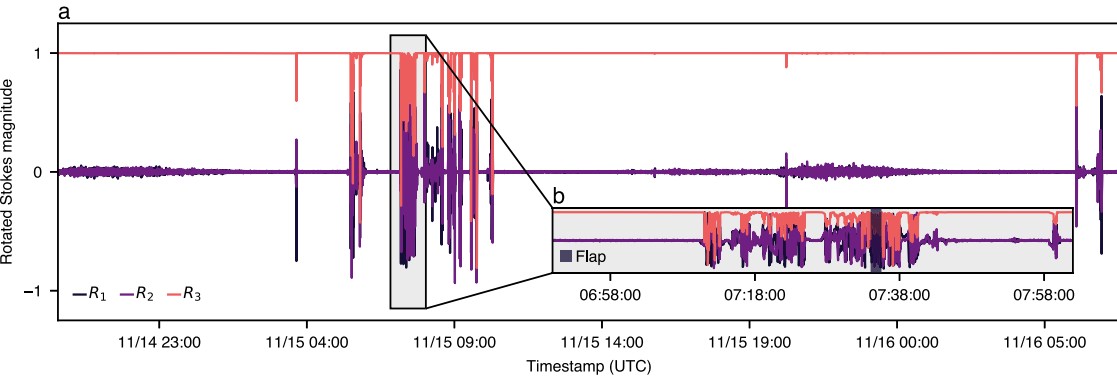

**Fig. 5 | State-of-polarization sensing foreshadows network downtime with a 20-min lead.** In this 30-h network snapshot captured between November 14th, 2022, and November 16th, 2022 (**a**), the network experienced an unplanned flap, i.e., a period of network downtime where no traffic flows, for 1.5 min (**b**). The rotated Stokes parameters began fluctuating 20 min prior, demonstrating the ability for state-of-polarization sensing to presage network downtime.

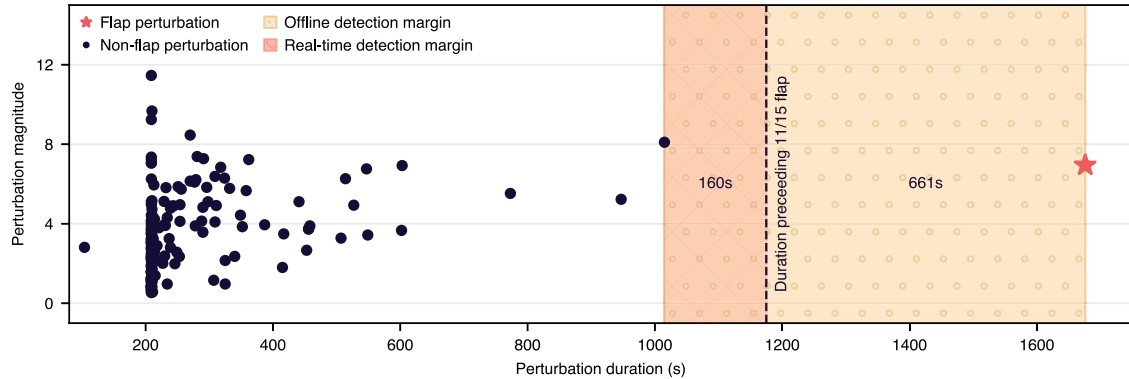

**Fig. 6 | Distribution of state-of-polarization perturbation windows automatically detected over the 85-day collection period.** We compute the duration of each perturbation window (x-axis) and the magnitude of state-of-polarization perturbations within each window (y-axis). Classifying purely on perturbation duration results in a wide decision margin of 661 s for offline detection and 160 s for online detection, enabling preemptive rerouting of network traffic before network flaps occur.

drift from previously stable points. Second, we compute the maximum value over a 60 s rolling window, then binarize the signal by comparing the rolling values to a fixed threshold. Third, we apply a second 150 s rolling window over the binarized values, taking the maximum to remove transient periods of SOP stability between otherwise high-intensity periods. The binarized signal now transitions from $0 \rightarrow 1$ when perturbations begin and from $1 \rightarrow 0$ when perturbations cease. The perturbation duration, $\Delta t$, is the time difference between adjacent $0 \rightarrow 1$ and $1 \rightarrow 0$ transitions, occurring at $t_0$ and $t_1$, respectively. To quantify the strength of SOP perturbations within each $\Delta t$, we define our perturbation magnitude as:

$$\frac{\sum_{t=t_0}^{t_1} \max\{|R_1(t)|, |R_2(t)|, |1 - R_3(t)|\}}{\Delta t} \qquad (4)$$

This metric considers the perturbation magnitude from all SOP channels, selects the maximum, and then normalizes the sum of strengths by $\Delta t$ to remove the bias towards longer perturbation windows.

As shown in Fig. 6, we apply this detection methodology to the 85 days of SOP measurements, resulting in 193 automatically detected perturbation windows. Immediately, we see that the non-flap perturbation windows have a mean $\Delta t = 275$ s and a mean magnitude of 3.5. Noticeably, the perturbation period containing our detected flap had a longer $\Delta t = 1676$ s (a 509% increase) and a comparably higher magnitude of 6.9 (a 97% increase). However, we also observe 17 non-flap perturbation windows (8.8%) with a magnitude larger than 6.9, indicating that the interplay between Stokes parameters is nonnegligible. We discuss an eigenvalue approach in the

"Discussion" section to address this observation. Regardless, using our recorded dataset, a naive linear classifier could pick a purely vertical boundary, i.e., classifying by perturbation duration alone, and still achieve a zero false positive rate. Taking this approach, we see a 661 s detection margin if we perform offline detection (i.e., waiting until the perturbation window has concluded) and a 160 s margin for real-time detection (i.e., we classify the perturbation window before it has ceased using the $\Delta t$ preceding the November 15th flap as a lower bound).

Given the scarcity of flap events within our 85-day collection window, we believe that further data collection is necessary to make explicit statistical claims on classification or forecasting ability in the presence of diverse flap events. We expect that classifying additional flaps will require more complex algorithms (e.g., logistic regression, perceptrons) or different input metrics with better theoretical roots. For example, our perturbation metric may be replaced with either (1) the distance from the point on the Poincaré sphere relative to the steady state position—which is obtained after rotating the Stokes parameters as discussed in the "Methods" section—or (2) the SOP change rate that sums the variations of the individual Stokes parameters. Regardless, this preliminary analysis demonstrates the possibility of real-time, automatic detection of flaps before they occur, with a sufficiently wide time margin to preemptively reroute traffic.

**Characterizing overall network health.** In addition to detecting acute instances of network downtime, we will now demonstrate how SOP sensing can characterize overall network health trends. As shown in Fig. 7, we compare the daily and hourly frequency of detected SOP/DAS events to globally recorded flap events reported by our internal telemetry

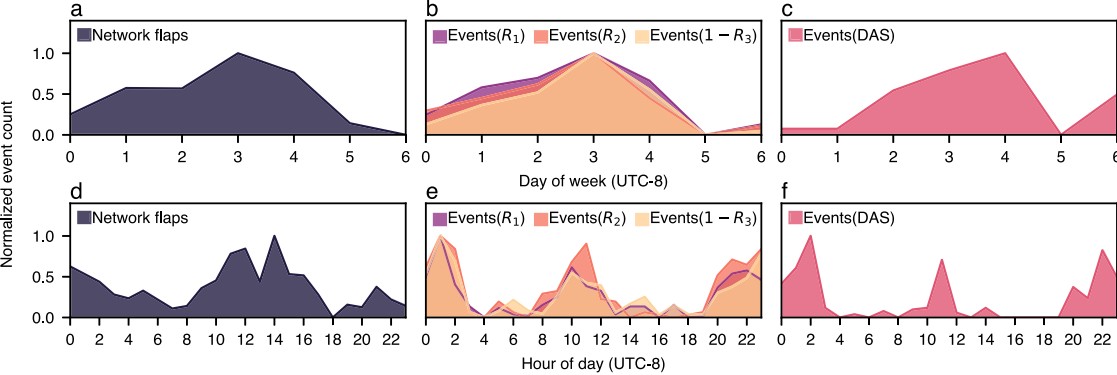

**Fig. 7 | Temporal distributions of network-wide flaps, state-of-polarization events, and distributed acoustic sensing events during the 3-month collection period. a–c** For daily distributions, 0 indicates Monday and 6 indicates Sunday. **a**, **d** Flaps are measured across the entire network. **b**, **e** Events are computed on our experimental fiber route using each rotated Stokes parameter. **c**, **f** Events are computed on our experimental fiber route using distributed acoustic sensing (DAS) signals. In all distributions, Stokes-based events have a comparable distribution to DAS-based events and network-wide flaps. Since network flaps are recorded from all fiber routes in the network, this indicates that polarization sensing along a single fiber can accurately model network-wide behavior.

monitoring system. Approximately 50% of samples originate from North America and 15% from Los Angeles. To classify an SOP/DAS signal as an event, we first compute the STA/LTA representation for each timestamp, perform min/max normalization to rescale the results between zero and one, and count the number of results above an empirical threshold (chosen to reduce the impact of false positives while still encompassing a wide range of stressor events that have the potential to affect fiber-optic network performance). For all three datasets, we limit events to those captured within the 3-month sensing interval. However, unlike the SOP and DAS measurements, which occur along the same fiber route in Los Angeles (and thus contain non-flap events), network flaps are collected along all fiber-optic routes in the network.

Observing Fig. 7, the distribution of daily network flaps and SOP events are nearly identical, rising slowly from Monday, peaking on Thursdays, and dropping until Sunday. Daily DAS events have a similar trend but show a peak on Fridays and a larger event count on Sundays. In terms of hourly distributions, all datasets follow the same general trend with local maxima at the early hours of the day/night and a midday peak. We also note that SOP sensing appears to surpass DAS in its ability to detect flaps, e.g., given the higher similarity between network flaps and SOP events between Sunday and Wednesday. We believe this may be due to SOP's nonlinear response to strain, which contributes more detected SOP events than DAS events and warrants further exploration in future work.

Our hypothesis that anthropic disturbances of fiber-optic cables primarily cause the majority of network outages is supported by the fact that the temporal distribution is concentrated around periods of high maintenance activity, i.e., middle of the night and midday. We note that the previously detected flap occurred around midnight UTC-8 (hour 0), aligning with the comparably high period of network flaps. Notably, the temporal similarity between network-wide flaps and SOP/DAS events along a single fiber-optic cable within the area demonstrates that the stressor events seemingly follow a predictable pattern that scales to the entire network. This insight can be leveraged to improve network-wide reliability by altering fiber maintenance trends, and the impact of such actions can be passively monitored with SOP sensing.

**Wide-scale seismic monitoring with SOP sensing**

Having demonstrated the potential of terrestrial SOP sensing, we now shift to a potentially transformative application: wide-scale seismic monitoring with unmodified fiber-optic networks. Dense seismic monitoring is widely accepted as being beneficial to improving earthquake early warning systems[62,63], the detection of low-magnitude seismic events[64] and improving seismic imaging[65]. Unfortunately, the cost of dedicated seismic stations and accompanying measurement devices severely limits coverage, leading

seismologists to consider low-cost sensor nodes[38] or portable devices, e.g., Raspberry Pis with MEMS accelerometers[64]. Although these approaches reduce the monetary overhead of dense seismic monitoring, they still require dedicated deployment. To mitigate this challenge, researchers have considered reusing vast fiber-optic networks for seismic monitoring, leveraging DAS and laser interferometry in both sub-sea[30,46,51,52] and terrestrial[8,27,29,66] environments. Unfortunately, as previously discussed, DAS and laser interferometry require dedicated hardware or ultra-stable laser sources to collect sensing measurements[36,67], posing a barrier to achieving widespread coverage.

Given this deployment challenge, an obvious choice is to consider SOP sensing of seismic events over unmodified fiber. In Fig. 8, we show the region's dedicated seismic monitoring stations[68] alongside publicly available fiber-optic network coverage[69] to demonstrate the potential to increase seismic monitoring coverage. Assuming each 50 km span of optical fiber (placed 1 km apart from adjacent fibers) could act as an independent seismic sensor, the roughly 5570 sq km area of fiber-optic cables could support over 111 fiber-based seismic stations. Including the 74 active locations of the Southern California Seismic Network, this more than doubles the number of independent local observations from 74 to 185, covering urban and remote areas (e.g., Santa Catalina Island).

Realizing this potential, however, is not without its challenges. First, in terms of coverage, access-layer fiber networks are typically too slow for coherent transponders, meaning the density of seismic sensing fiber routes would vary depending on the region. However, future backhaul to 5G towers are expected to utilize coherent transponders, thereby improving seismic sensing coverage. Second, SOP sensing of seismic activity is still in its infancy compared to alternative methodologies (e.g., DAS) but has recently shown promising results in sub-sea environments[46,51]. Third, although SOP seismic sensing has been hypothesized to work on terrestrial networks[19], it has not been demonstrated outside of laboratory environments, given the stronger noise floor of terrestrial vs. sub-sea environments. Regardless, given our unique access to long-term SOP measurements in the highly active seismic region of Southern California, we perform a brute-force search over the magnitude ≥Ml 1 earthquakes ($n = 475$) occurring within a 100 km radius of our optical fiber (comparable to the sensing range of other earthquake early warning systems and larger than the typically reported blind zone radius[70,71]) and record the expected arrival of the $P$ and $S$ body waves using the IRISWS traveltime API. For candidate earthquakes with SOP perturbations following the phase arrivals, we query the corresponding seismograms from the nearby ($\approx$20 km) CI PASC station[68] using the IRISWS timeseries API.

Across all analyzed earthquakes, we identify three candidates with similar SOP perturbations following the arrival of seismic waves. Figure 9

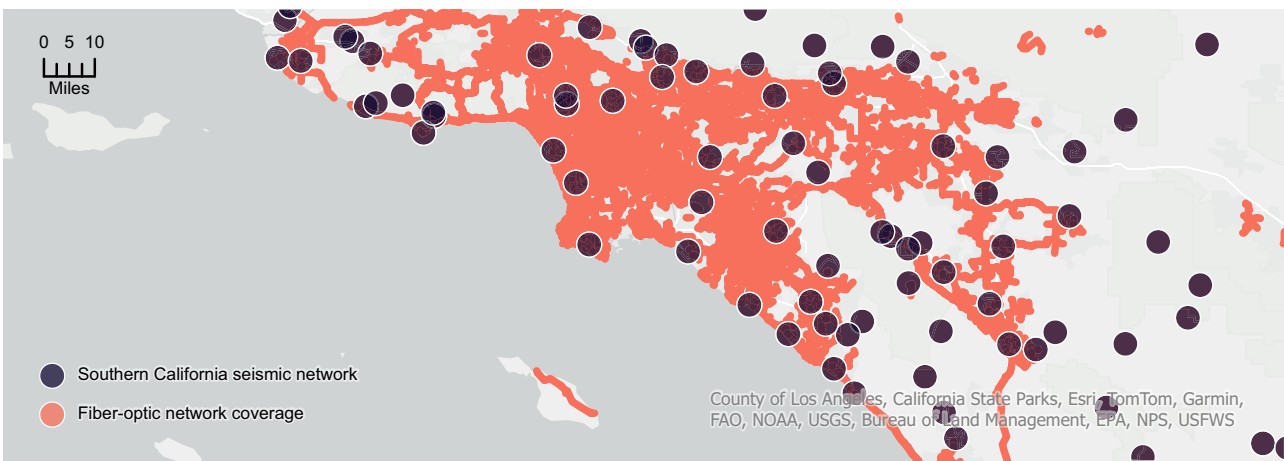

**Fig. 8 | Southern California's fiber-optic coverage plotted alongside the region's seismic monitoring stations.** Leveraging existing fiber-optic networks for passive seismic sensing can more than double the number of independent local observations, from 74 to 185, in both urban and rural areas.

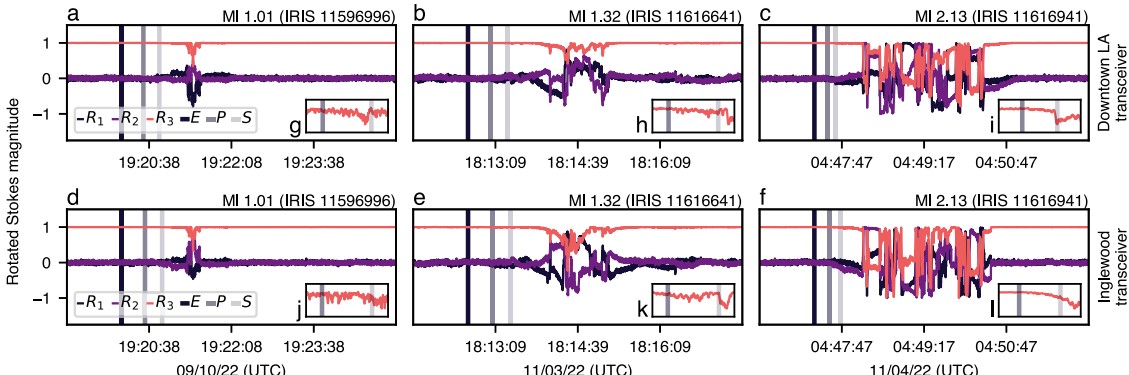

**Fig. 9 | Rotated Stokes magnitudes following three earthquakes.** The origin time $E$ (given by the seismogram stations) and the estimated $P$ and $S$ phase arrivals to the Downtown Los Angeles and Inglewood optical transceivers are plotted as vertical lines. **a–f** The rotated Stokes parameters slowly drift following the arrival of seismic waves, with drastic perturbations occurring ≈30 s after $S$ arrival. **g–l** demonstrate the increasing perturbations of $R_3$ following the arrival of the seismic waves.

plots the rotated Stokes parameters measured at both ends of the fiber-optic route. During these periods, no fiber maintenance was scheduled, and no reported network downtime was observed. In all three cases, the rotated Stokes parameters begin to drift following the arrival of $P$ waves, which is further amplified after the arrival of $S$ waves. The insets of Fig. 9 show the increasing perturbations of $R_3$ following the arrival of these seismic waves. Approximately 30 s after the drift begins, the rotated Stokes parameters wildly fluctuate for a short duration (1–4 min) before returning to a steady state. Notably, the relatively minor perturbations following the arrival of $P$ and $S$ waves appears to indicate that SOP is more sensitive to the surface wave packet of the earthquake's wavefront (which arrives after the body waves). Assuming a wave speed between 1.5 km s$^{-1}$ and 3 km s$^{-1}$, the more robust perturbations are aligned with the surface wave's theoretical arrival time to the optical fiber.

Despite these promising observations, it is worth noting that only a small portion (<1%) of the analyzed earthquakes had any meaningful SOP perturbations following the wavefront arrival. Still, all followed the above pattern if perturbations were present. We attribute this lack of sensitivity to two potential phenomena. First, it has previously been shown that DAS's sensitivity to seismic activity highly depends on the type of cable where the fiber is placed and the type of coupling to the ground[26]. Second, the laboratory analysis conducted in ref. [19] indicates that pure polarization-based sensing of earthquakes is limited to specific types of seismic events. The "Discussion" section examines the potential to improve sensitivity.

Taking a deeper dive, we now scrutinize the largest magnitude earthquake observed with matching SOP perturbations (IRIS 11616941) and

correlate the SOP signal with DAS and a nearby seismic station (Fig. 10). Analyzing the time derivative of the BHZ channel, we observe a distinct peak when the $P$ waves arrive at the station, followed by additional perturbations upon arrival of the $S$ waves. Regarding SOP, the rotated Stokes parameters begin to drift as the $P$ and $S$ waves arrive at each optical transceiver, followed by strong SOP perturbations ≈30 s later. The DAS waterfall shows a similar pattern, beginning with a large strain rate at an aerial section of the fiber (channel 2600) after the arrival of the $P/S$ waves and then strain perturbations across all channels ≈30 s later.

Given that the high-strain rate begins at an aerial section of the fiber, we hypothesize that aerial sections sway during seismic activity due to vibrations traveling through the coupling structure (e.g., wooden poles or concrete structures[72]), resulting in resonant SOP perturbations that produce similar spectral signatures as moving fiber spools (see Fig. 2). To explore this theory using the data recorded during our measurement campaign, we analyze the impact of a similar agitation event on an aerial fiber section, namely high-intensity wind gusts. Using the Open-Meteo API[73–76], we query historical wind gust measurements localized to an aerial section of the fiber and measured 10 m in the air. As shown in Supplementary Fig. 2, we see a sharp decrease in the $S_1$ and $S_2$ Stokes parameters when the wind gusts exceed 40 km h$^{-1}$, followed by a return to their prior steady state as the wind gusts decline. We also query historical wind data at the time of phase arrival to rule out the possibility of coincidental wind gusts impacting the previously identified seismic perturbations. Across all three scenarios, we observe 38% smaller wind speeds and 29% smaller wind gusts than the environment depicted in Supplementary Fig. 2.

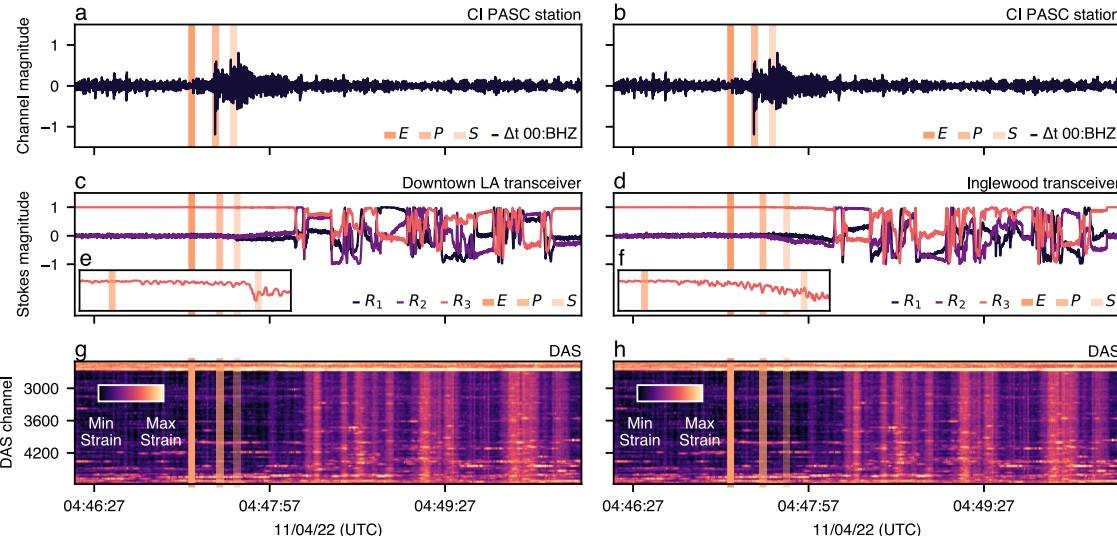

**Fig. 10 | Ground truth seismic station waveforms, fluctuating Stokes parameters, and distributed acoustic sensing waterfall during the IRIS 11616941 earthquake.** In all panels, the earthquake's origin time $E$ is given by the CI PASC seismogram station[68]. **a, b** plot the time derivative of the seismic station's 00:BHZ channel and estimated $P$ and $S$ phase arrivals. **c, d** show the fluctuating Stokes parameters after estimated phase arrivals at both optical transceivers. **e, f** zoom in on the increasing perturbations of $R_3$ following the arrival of the seismic waves. **g, h** show the corresponding distributed acoustic (DAS) sensing waterfall measured by the optical interrogator. Notably, the Stokes parameters begin to drift as DAS registers a high-strain rate on an aerial fiber section (DAS channel 2600).

Having shown the feasibility of terrestrial SOP seismic sensing, we now discuss how to leverage these observations. In terms of increasing seismic monitoring coverage for scientific pursuits, SOP sensing over optical fiber is an exciting option. With a potential 150% increase in independent seismic observations, the capability to continuously monitor environmental changes and human activity is more than doubled in Southern California. In terms of aiding in earthquake early warning systems, careful designs must be considered, as our data indicates that SOP sensing is most sensitive to surface wavefronts. This contrasts traditional early warning systems, which detect the arrival of the fast-moving $P$ waves to improve response time. However, since the increased noise floor of urban environments makes $P$ wave sensing challenging[77], dense monitoring of surface waves is an attractive alternative. Furthermore, as outlined in the "Methods" section, reliable characterization of SOP waveforms corresponding to different stressor events (e.g., seismic activity, wind gusts swaying aerial fiber sections, fiber maintenance activities) is still an open challenge. In the context of seismic monitoring, we hypothesize that leveraging more advanced machine learning techniques (e.g., ref. 78) offers the potential to reliably distinguish between received waveforms. When combined with the lightning-fast communication of the underlying fiber-optic network, this smart-grid approach may leverage many optical links far away from populated areas, which can detect incoming earthquakes and save seconds for early warning systems.

## Discussion

In this work, we have demonstrated the potential of SOP sensing when applied to terrestrial fiber-optic networks. First, we validated the accuracy of SOP sensing with an 85-day correlation to DAS ground truth, showing a strong cross-correlation between the two sensing methodologies. Given this high accuracy, we next evaluated the potential of SOP sensing on two applications. First, we demonstrated the ability to automatically detect periods of network downtime, i.e., flaps, with a real-time 160 s detection margin. Second, we showed that SOP sensing over a single fiber can provide a snapshot of network-wide health trends. Third, we explored the possibility of leveraging existing fiber-optic networks for terrestrial seismic sensing.

It is worth noting that throughout this study, we have analyzed the perturbations of the scalar Stokes parameters in isolation and not the Stokes vector as a whole or the interplay between its components. A recent work[19] has proposed an alternative eigenvalue approach to SOP sensing, measuring

the polarization rotation matrix instead of individual scalar components. In general, vector analysis (which can be accomplished using Jones calculus[79]) enables sensing the dynamic interactions between the Stokes parameters that are otherwise ignored with scalar analysis. However, it was also shown in ref. 13 that when SOP perturbations are small enough, the scalar Stokes parameters give the same result as two of the three components of the rotation vector characterizing the Jones matrix in Stokes space. In this work, the deviations of $S_1$ and $S_2$ from the stationary point (i.e., North pole as described in the "Methods" section) are equivalent (for minor deviations) to the fluctuations of the component of the rotation vector in Stokes space that is orthogonal to the input Stokes vector. Looking to future applications, the optimal analysis methodology depends on the expected SOP characteristics of the intended sensing application, and we leave evaluating these more complex situations to future work.

Moving beyond our evaluation, we believe that SOP sensing should be further explored in the context of current terrestrial DAS and interferometric applications. Without question, DAS and interferometric sensing are tremendously powerful tools for detecting various fiber stressor events. Specific to DAS, because of its distributed nature, it can paint a comprehensive picture of strain along different spatial sections of fiber. Such information can be used to derive information about the conditions of the physical placement of the fiber (e.g., aerial vs. terrestrial) and the dynamics of its surrounding environment. Enabling such spatial localization with SOP sensing would further increase its practicality as a viable DAS alternative when a nonlinear response to strain and lower sensitivity is acceptable. Some recent works have explored bringing localization to SOP sensing through numerous methods[19,41–44], and offer an opportunity to be evaluated in real-world terrestrial environments.

Regarding immediate adoption, we believe it is important to note our unique access to the internal registers of our coherent transponders, which enabled the SOP sensing used throughout this study. Many transceivers from different vendors do not inherently expose polarization data due to the closed-source nature of the internal DSPs. This, in turn, has led researchers to consider alternative means of sampling SOP changes before being processed by the transceiver. For example, in ref. 44, the authors leverage polarizing beamsplitters and polarimeters to measure SOP directly from the intensity-modulated direct detected (IMDD) data signals. This design offers an exciting opportunity for SOP sensing in the absence of internal DSP access, as it can be implemented on either the (1) optical supervisory

channels present at many amplification sites or (2) transceivers in access network segments that still largely leverage IMDD transceivers. Regardless, we remain hopeful that transceiver vendors will modify their hardware/ software to enable direct SOP access, enabling ubiquitous access to polarization waveforms without any deployment overhead.

Looking to the future, additional networking and seismic applications should continue to be explored. For networking applications, machine learning techniques can be applied to improve the detection accuracy of diverse flap perturbations. In terms of seismic monitoring, leveraging multiple fiber spans across a wide area can begin efforts toward developing a fiber-based earthquake early warning system. Above all, we believe that the ability of SOP sensing to be deployed on unmodified fiber-optic networks makes it a potent sensing tool, and we expect future applications will continue to exploit this power for transformative purposes.

## Data availability
Underlying SOP measurements, DAS measurements, and aggregate network information are available from the corresponding author upon reasonable request and with the permission of Google. The corresponding author should be contacted via email with a justification for the request, the intended usage of the data, and any other relevant information. Seismic and wind data are publicly available via the IRISWS timeseries API and Open-Meteo API. Fiber-optic coverage and seismic station details are available at the previously referenced locations[68,69].

## Code availability
Analysis code is available from the corresponding author upon reasonable request and with the permission of Google. The corresponding author should be contacted via email with a justification for the request, the intended usage of the code, and any other relevant information. All maps were generated using the ArcGIS Pro software by Esri.

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

## Acknowledgements

We sincerely thank Google for providing access to DWDM transport, DAS equipment, dedicated dark fiber, and the sensing infrastructure used in our experiments. We thank Tad Hofmeister, Jorge Castillo, Mattia Cantono, and Valey Kamalov for their invaluable feedback and domain expertise. We also thank our reviewers for their insightful comments. C.J.C. discloses support for the research of this work from the National Science Foundation (GRFP-1840344 and DGE-2036197). Any opinions, findings, conclusions, or recommendations expressed in this material are those of the authors and do not necessarily reflect those of the funding agencies or others. The IRIS Data Services facilities, specifically the IRIS Data Management Center, were used to access waveforms, related metadata, and derived products used in this study. IRIS Data Services are funded through the Seismological Facilities for the Advancement of Geoscience (SAGE) Award of the National Science Foundation under Cooperative Support Agreement EAR-1851048.

## Author contributions

Charles J. Carver designed the experiments, collected all data, performed the analyses, and prepared the manuscript's text and figures. Xia Zhou helped organize and review the final manuscript.

## Competing interests

The authors declare no competing interests.
