## [Peer Review File · Communications Engineering]

Web links to the author's journal account have been redacted from the decision letters as indicated to maintain confidentiality.

Decision letter and referee reports: first round

12th March 2024

Dear Mr Carver,

Your manuscript entitled "Polarization sensing of network health and seismic activity over a live terrestrial fiber-optic cable" has now been seen by 3 referees. You will see from their comments below that while they find your work of interest, some important points are raised. We are interested in the possibility of publishing your study in Communications Engineering, but would like to consider your response to these concerns in the form of a revised manuscript before we make a final decision on publication.

We therefore invite you to revise and resubmit your manuscript, taking into account the points raised. Please highlight all changes in the manuscript text file. Please do not hesitate to contact me if you have any questions or would like to discuss these revisions further.

In the event that your manuscript is accepted, we will provide detailed guidance on our journal policies and formatting. In order to save time later, you may wish to ensure that the manuscript broadly complies with our house style at this stage. See our style and formatting guide for details: (<https://www.nature.com/documents/commsj-phys-style-formatting-guide-accept.pdf>) and checklist (<https://www.nature.com/documents/commsj-phys-style-formatting-checklist-article.pdf>)

To improve the quality of methods and statistics reporting in our papers, we are now asking all authors to complete an editorial policy checklist that verifies compliance with all required editorial policies. Please ensure that the checklist is completed and uploaded with your revised article. All points on the policy checklist must be addressed; if needed, please revise your manuscript in response to these points. Please note that this form is a dynamic 'smart pdf' and must therefore be downloaded and completed in Adobe Reader. Clicking this link will download a zip file containing the pdf.

Data and Code Availability

If you have not already done so, please now disclose to the editors any restrictions on the availability of materials or information central to the main claims of the manuscript. This includes data and code which are central to the key conclusions. Nature Portfolio policies for sharing of research materials and data can be seen here.

All Communications Engineering manuscripts must include a section titled "Data Availability" at the end of the Methods section or main text (if no Methods). More information on this policy, and a list of examples, is available at <http://www.nature.com/authors/policies/data/data-availability-statements-data-citations.pdf>. For more details on our open data policies, please refer to our data policies at <http://www.nature.com/authors/policies/availability.html>.

For all studies using custom code or mathematical algorithm that is deemed central to the conclusions, a statement must be included under the heading "Code Availability", indicating whether and how the code or algorithm can be accessed, including any restrictions to access. Code availability statements should be provided as a separate section after the data availability statement but before the references.

Ideally, authors of papers reporting a central advance of code and software should complete the following Checklist to aid with reviewing the manuscript's key contribution: Checklist for papers with a central advance of new custom code or software.

Please use the following link to submit your revised manuscript, point-by-point response to the referees' comments (which should be in a separate document to any cover letter) and any

Decision letter and referee reports: first round

completed checklist:

[Link Redacted]

We hope to receive your revised paper within six weeks; please let us know if you aren't able to submit it within this time so that we can discuss how best to proceed. If we don't hear from you, and the revision process takes significantly longer, we will close your file. In this event, we will still be happy to reconsider your paper at a later date, as long as nothing similar has been accepted for publication at Communications Engineering or published elsewhere in the meantime.

Sincerely,

Anastasiia Vasylychenkova, PhD
Associate Editor
on behalf of

Carmine Galasso, PhD
Editorial Board Member
Communications Engineering
orcid.org/0000-0001-5445-4911

Reviewers' comments:

Reviewer #1 (Remarks to the Author):

Review of "Polarization sensing of network health and seismic activity over a live terrestrial fiber-optic cable," by Charles J. Carver and Xia Zhou

The paper is a welcomed contribution that responds to the need of a comparison between DAS and SOP in monitoring fiber perturbations of various types, either of anthropic origin or of seismic nature. I endorse publication. A few suggestions to improve, to my opinion, the quality of the manuscript.

Page 4: If it is possible, it would be useful to give the reader a concise qualitative explanation of what short-term-average-long-term-average (STA/LTA) does.

Instead of using the metric of Eq. (1), which has little theoretical grounds, why don't use the distance of the point on the Poincare sphere from the steady state position (the value obtained by averaging, which is set to the North pole)? Since the Stokes parameters lie on the unit sphere $S_1^2 + S_2^2 + S_3^2 = 1$, this distance can be obtained by using either $S_1^2 + S_2^2$, or $1 - S_3^2$. While I am not expecting that the authors redo their analysis with the new metrics, I leave this as a suggestion for future studies.

It is shown that only about 1% of seismic events are recorded using SOP. Rather than attributing this lack of sensitivity to the characteristic of the seismic perturbations, the different sensitivity may more likely be caused by the different type of cable where the fiber is placed (jelly-filled loose tube, or other types), and by the different coupling of the cable itself with the ground. It would be interesting to know the characteristics of the optical cables that were successful in the recording and their coupling with the ground. It is indeed known that also DAS sensitivity is strongly affected by the cable characteristic and by the coupling with the environment.

Concerning the use of the full Jones matrix (in the Discussion session), in ref. [A] below it was shown that the use of the SOP vector gives the same result (in the regime where the SOP perturbation is small enough to be useful) of the use of two of the three components of the rotation vector fully characterizing the Jones matrix in Stokes space. Indeed, the rotated Stokes parameters used in this paper, namely the deviations of S_1 and S_2 from the stationary point

Decision letter and referee reports: first round

when the Poincare sphere is rotated such that the stationary point is the North pole, are equivalent for small deviations, to the fluctuations of the component of the rotation vector in Stokes space orthogonal to the input Stokes vector, when a reference frame rotating with the static birefringence is used (see again ref. [A] below).

Antonio Mecozzi

[A] Antonio Mecozzi, Cristian Antonelli, Mikael Mazur, Nicolas Fontaine, Haoshuo Chen, Lauren Dallachiesa, and Roland Ryf, "Use of Optical Coherent Detection for Environmental Sensing," *J. Lightwave Technol.* 41, 3350-3357 (2023).

[B] Costa, L., Varughese, S., Mertz, P. et al. "Localization of seismic waves with submarine fiber optics using polarization-only measurements," *Commun Eng* 2, 86 (2023).
<https://doi.org/10.1038/s44172-023-00138-4>.

Reviewer #2 (Remarks to the Author):

Integrated SOP sensing has been demonstrated in several contexts in recent years, and has quickly garnered attention, making this work timely. The technique used is not novel, but the main distinction of this work is the scale of its measurement campaign. The authors also propose some interesting applications regarding the improvement of network reliability, which are not commonly mentioned in the literature.

I believe this work is an interesting case-study of integrated SOP as a cost-effective and readily implementable fiber optic sensing technique and may be of interest to a broad audience, and am willing to recommend it pending some alterations to the text.

As a general suggestion, I think that in some instances the authors were slightly liberal in stating the impact of their work. For example: "In the following sections, we will demonstrate two high-impact applications that SOP sensing is primed to support: strengthening network robustness and wide-scale seismic monitoring.", "...tremendous power of SOP sensing", "..Given this high accuracy, we next evaluated the transformative potential of SOP sensing on two high-impact applications."

The impact of the method and its applications should be determined by its adoption after the fact. I think the authors could also reduce the amount of hyperbolic language.

Specific comments:

Introduction:

- In the first paragraph of the introduction, ref. 4 is cited twice.

- "In terms of its technical overhead, because of the larger data volume associated with DAS's high sampling-rate, DAS demands powerful computational, storage, and processing capabilities that are generally only available in high-cost systems [43, 44]."

The "larger data volume" here is being presented as a disadvantage for DAS. In my view, this is an odd position: DAS has a maximum sampling rate (limited by the fiber length), but not a minimum sampling rate. In fact, it can be argued that the data-per-channel-per-acquisition of DAS is lower, since in this case for a single channel (the full integrated span of fiber), there are 3 Stokes vectors to be captured, while DAS captures directly a temperature/strain/phase/frequency shift value per sample. The high sampling rates and spatial resolution are the cause of the increased demands for storage, but all of these can be tuned to the same level as the SOP technique described.

DAS has several advantages over SOP sensing but is a high-cost technique, typically demands high power launched into the fiber, is incompatible with in-line repeaters with isolators, and demands specialized narrow-linewidth lasers - these factors sufficiently justify the consideration of SOP sensing.

Decision letter and referee reports: first round

- "SOP sensing has emerged as a viable DAS alternative. Unlike DAS and interferometric systems, SOP sensing analyzes the integrated polarization changes of the modulated light traversing through traffic-carrying optical fibers [14, 15, 16]." Some recent (and not-so-recent) works have used SOP measurements without doing the full integrated fiber span. It may be more accurate to describe SOP sensing's capabilities in both localized and integrated applications, as demonstrated with P-OTDR [1], the recent publication by Fatih et al. (already referenced by the authors), and other methods like [2] and [3].

- When listing contemporary sensing methods, there is no mention of inelastic scattering based techniques - I don't think the list here needs to be exhaustive, but it seems like an oversight seeing as techniques like BOTDA and ROTDR are among the most common fiber-optic sensing methods, and likely more commonplace than SOP measurements.

Results and Discussion:

- Figure 2, I believe, should be referenced in section 2.1. It is only referenced much later in the text.

Also, despite the obvious effects on the optical fibers, there seems to be no comment on ability to discriminate between different events.

I don't think this is necessarily possible, unless the type of perturbations experienced is particularly constrained to predetermined type (for example, bending in [1]), but since the nonlinearity to strain is a known challenge of SOP methods over other strain-measuring techniques such as DAS, microwave, or Brillouin, it should be clearly mentioned in the text.

- It took me a while to understand that the correlations in figure 4 are correlations of the triggers/events, and not the acquired time-series themselves. Again, SOP signals are not linear with the perturbation, and I believe this should be more evident, when talking about the STA/LTA algorithm in section 2.2. Perhaps the authors should make this more clear in the figure caption, to prevent misunderstanding the data.

4. Regarding flap detection: I thought this is an interesting application that I have not seen previously in the literature, and indeed may be a good fit since it might not require linearity and spatial resolution (such as seismology for example), and benefits from the ease-of-access of integrated SOP methods.

Nonetheless, I think that one of the weaknesses of the presented data is that only one flap event occurred (figure 6), which makes it hard to generate a statistically significant model determining the correct thresholds for predicting the flap.

It is true that this event was accompanied by an unprecedented perturbation duration, but correlation does not always imply causation, and with only one event it is hard to make any statistical claims about classification or forecasting ability. Can the authors comment on this? Is there any possible analysis of false-positive/false-negative rate that can be made?

The data in figure 7 seems to support the fact that SOP can even surpass DAS in its ability to detect flaps, going by distribution of recorded events. Is there any reason for this?

- In Figure 9, it seems that the authors only report earthquakes very close to the fibers, and of low magnitudes. Given the length of the measurement campaign, has there been any demonstration of measurement of larger earthquakes, at farther distances? That might be a more interesting scenario as a case-study for early warning, as this application is predicated on the ability to sense P-waves.

Related to this, I think the following statement in the discussion is not necessarily true: "Third, we explored the possibility of leveraging existing fiber-optic networks for terrestrial seismic sensing, demonstrating the potential to increase Southern California's seismic monitoring coverage by 150%. Combined with a thorough analysis elucidating the impact of seismic waves on terrestrial

Decision letter and referee reports: first round

fiber, we have demonstrated how optical networks can aid in environmental monitoring and earthquake early warning systems."

Indeed, optical networks can benefit environmental monitoring, but for earthquake early warning, using SOP, I don't think the data presented in this paper supports that case.

Discussion:

- "Enabling such spatial localization with SOP sensing would increase its practicality and transform it into a full-fledged DAS replacement.". Some works (see refs [1],[2],[3] and Fatih et al. work) have shown spatial localization using SOP sensing (although not in terrestrial networks). More importantly, SOP is not set-up to be a DAS replacement, even with localization: DAS has a linear response to applied strain, and seems to be much more sensitive, even with comparably short gauge lengths.

[1] - Distributed polarimetric measurements for optical fiber sensing, Luca Palmieri, Optical Fiber Technology, 19, 6, 2013, 720-728

[2] - Costa, L., Varughese, S., Mertz, P. et al. Localization of seismic waves with submarine fiber optics using polarization-only measurements. Commun Eng 2, 86 (2023).

[3] - A. Galtarossa, D. Grosso, L. Palmieri and L. Schenato, "Reflectometric Characterization of Hinges in Optical Fiber Links," in IEEE Photonics Technology Letters, vol. 20, no. 10, pp. 854-856, May15, 2008

Reviewer #3 (Remarks to the Author):

Dear Authors and Editor,

thank you for having submitted the your valuable work to Nature Communications Engineering. The paper is focused on the exploitation of the terrestrial telecommunications networks infrastructure for environmental sensing leveraging on state of polarization (SOP) monitoring obtained by deployed transceivers. This idea is gaining interest not only in the telecommunications/optical community, but also in the earth sciences community as its implementation would, at the same time, double the optical networks value offering additional services and open the doors for immense dataset to the geophysics people. Moreover, while several papers have been published on this topic, limited to subsea optical links, less work has been done on terrestrial networks. In the reviewer's opinion this already represents an important selling point for this work. Indeed, terrestrial sensing offers several advantages (and disadvantages) w.r.t. subsea sensing. Subsea communications are commonly point-to-point-links and SOP can be sampled only at transmitter/receiver sites and it's very difficult to add further devices and SOP monitoring devices as they are placed underwater.

Terrestrial networks are instead usually arranged in several topologies as ring or mesh, and it is easier to access and upgrade transceivers, add/drop sites or add additional sensors. The possibility to have more sensing points across the network can counterbalance the fact that SOP sensing provides integrated measurements in space w.r.t. to DAS and cover wider areas. This is clearly stated in the paper, and I really appreciate the authors to clearly show to the reader a rough estimation of how much the traditional sensor network considered can be enlarged integrating SOP sensing. On the other side, terrestrial SOP sensing is afflicted by a much larger amount of noise w.r.t. the underwater scenario, generated by several anthropic activities, especially in a largely crowded area.

In this context, a paper like this, based on a solid long term observation (86 days) of real deployed devices compared to equivalent parallel fiber DAS observation, surely deserves publication on this journal. However, apart from some minor aspects that can be adjusted or clarified, I have some concerns about the generality of the main findings, although I understand that such experimental work on a such novel subject may not always scale to find general rules. I will discuss the main points in the following:

Decision letter and referee reports: first round

- I would move section 4 straight after the Introduction section. This is substantially a section where the authors explain the data retrieval and post-processing methodology. This would make much clearer to the reader the following discussion. Also, supplementary figures can be included in the main paper flow for better readability. Also, if not covered by NDA, the author may report the used transceiver model and the characteristics in terms of sampling rate and the accuracy of the Stokes vector measurement (if available) provided.
- In section 2 the authors discuss how different fiber stressors (bending, moving, etc.) impact Stokes vector and they report one evolution per type of event. It is known that polarization evolution depends on the stochastic realization of the fiber birefringence. My biggest concern here is about the repeatability and distinguishability of those typical waveform. In my opinion the authors should clarify to what extent the obtained waveform for a specific stressor can be more or less strictly identified by some features of typical evolution and how much those waveforms are distinguishable among the different stressors, for example, by means of crosscorrelation analysis.
- In section 2.2 is not clear what Q1, Q3 values mean. The whole explanation about the crosscorrelation lags appears in general not clear. Also, the reference base of time used should be clarified. From Supplementary Fig.1 caption it seems that the SOP perturbation is detected prior to DAS (-5/-6 seconds) for the same event and that this lag increases due to timestamp drifts. Need for precise time synchronization in data network sensing is a known issue.
- At the end of section 2.2, its not necessarily true that SOP sensing has necessarily low data storage requirements. Applications outside the real time early warning service, such as recording SOP waveform for offline geophysical studies, may require a lot storage depending on also on the SOP sampling rate.
- There has been some investigation on the localization using SOP sensing by the group of E.Virgillito, R. Bratovich et al. [1], even when coherent transceivers are not available. This can be done by performing SOP sensing on opposite direction. You may cite [1] to mention this.
- In section 2.3 the authors take over on the usage of SOP sensing for network reliability. In this case I would doubt the generality of the approach given that you observed 17 non-flap events with magnitude larger than 6.9. As the authors mention, the interplay between the Stokes can be significant. To this aim, I would suggest considering an aggregate metric such as the SOP change rate, which sums the variations of S1, S2, S3, other than more complex classifiers such as ML-based as suggested by the authors.
- Although the reporting of the single flap event is quite interesting, it seems too limited to derive thresholds for preemptive rerouting. From section 2.3.2 indeed is not clear if in the temporal distribution also non-flap events are included. If only events corresponding to an effective network flap are reported, comparing the SOP waveform characteristic to the single event reported in section 2.3.1 would be interesting. Also, in section 2.3.2, it would be useful to mention the source for the global flaps data.
- Section 2.4 is then demanded to the analysis of the collected data for seismic monitoring. Regarding the scalability of the SOP monitoring approach on coherent transceivers, practical issues on accessing Stokes vector should be mentioned: while the authors had the opportunity to access the internal registers of the coherent transceivers, most of these transceivers from different vendors do not expose this data due to the closed source nature of the DSP, although surely available at low level, and it is not straightforward for whatever network operator to obtain access to them after deploying in production network due to constraints imposed by transceiver vendors and technical limitations. In [1] the authors proposed an approach to overcome these limitations: SOP data can be monitored from intensity modulated – direct detected (IMDD) data signals using cheaper SOP monitoring devices based on polarization beams splitters. Although this requires additional devices and expenditures, this approach can leverage on a wide number of IMDD sources, such as optical supervisory channels present at many amplification sites or transceiver in the access network segments which still largely operated with IMDD transceivers.
- I suppose that origin time E in Fig.9/10 is given by the seismograms station. Please clarify this.
- The authors state that the aerial section of the fiber act as an amplifier for the incoming P/S waves and support this statement with discussion based on Supplementary Fig.2. I agree that

Decision letter and referee reports: first round

strong fiber shaking due to wind gusts would trigger large SOP oscillations. However, I am not really convinced that such oscillations can amplify SOP oscillations due to seismic waves, especially since fiber aerial sections are not directly coupled to the ground motion. Moreover, independently on the sampling rate of the internal DSP polarimeter offered to the final user, the actual sampling rate used for data signal reception is usually of the same order of magnitude of the optical channel symbol rate, which should be between 32 and 96 GBaud (10^9 Hz order) in state of the art transceivers, to successfully track data signal. Hence, tolerance to SOP oscillation rate should be typically far slower than the sampling rate, otherwise the receiver would lose the lock on the received channel leading to a potential network flap. Hence, this would suggest that strong SOP oscillations observed may be linked to fiber oscillations caused by wind, which show their effect on the integrated fiber length and that the amplification mechanism needs further evidence from the authors.

- This point leads to the more general topic of how SOP oscillations caused by seismic waves can be distinguished from other sources, such as wind gusts for aerial sections or possible fiber maintenance activity, which is my main concern about the presented claims. I am quite surprised to not see a larger amount of noise caused by other anthropic activities in the P wave segments considering that the optical link under observation is deployed in a highly populated area, which may also partially explain why among the 478 candidate earthquakes only less than 1% have shown meaningful SOP perturbations. Possibly, machine learning based approaches as in [2] may improve the situation. In this context, I agree that surface wave sensing is still an attractive opportunity if used in a smart grid context where many optical links, farther from highly populated areas, can be used simultaneously to detect incoming earthquake and save precious seconds for early warning.

Best Regards, The Reviewer

[1] E. Virgillito et al., "Detection and Localization of Metropolitan Anthropic Activities by SOP Monitoring of IM-DD Optical Data Channels," in 2023 International Conference on Photonics in Switching and Computing (PSC), Sep. 2023, pp. 1–3. doi: 10.1109/PSC57974.2023.10297183.

[2] Z. Li, M.-A. Meier, E. Hauksson, Z. Zhan, and J. Andrews, "Machine Learning Seismic Wave Discrimination: Application to Earthquake Early Warning," *Geophysical Research Letters*, vol. 45, no. 10, pp. 4773–4779, 2018, doi: 10.1029/2018GL077870.

We would like to thank each of our reviewers for taking the time to read and comment on our manuscript. We appreciate your kind words on the value of our work and your extremely helpful comments. We have addressed each of them below to the best of our ability, and believe these suggestions have resulted in an improved manuscript.

Reviewer #1

Review of "Polarization sensing of network health and seismic activity over a live terrestrial fiber-optic cable," by Charles J. Carver and Xia Zhou

The paper is a welcomed contribution that responds to the need of a comparison between DAS and SOP in monitoring fiber perturbations of various types, either of anthropic origin or of seismic nature. I endorse publication. A few suggestions to improve, to my opinion, the quality of the manuscript.

Page 4: If it is possible, it would be useful to give the reader a concise qualitative explanation of what short-term-average-long-term-average (STA/LTA) does.

We have added a one-line quantitative explanation of STA/LTA. We have also moved the Methods section after the Introduction to give all foundational information earlier rather than later.

Instead of using the metric of Eq. (1), which has little theoretical grounds, why don't use the distance of the point on the Poincare sphere from the steady state position (the value obtained by averaging, which is set to the North pole)? Since the Stokes parameters lie on the unit sphere $S_1^2 + S_2^2 + S_3^2 = 1$, this distance can be obtained by using either $S_1^2 + S_2^2$, or $1 - S_3^2$. While I am not expecting that the authors redo their analysis with the new metrics, I leave this as a suggestion for future studies.

We agree with your observation and included your suggested flap detector metric. We have also added a note that additional metrics can be considered depending on the intended application.

It is shown that only about 1% of seismic events are recorded using SOP. Rather than attributing this lack of sensitivity to the characteristic of the seismic perturbations, the different sensitivity may more likely be caused by the different type of cable where the fiber is placed (jelly-filled loose tube, or other types), and by the different coupling of the cable itself with the ground. It would be interesting to know the characteristics of the optical cables that were successful in the recording and their coupling with the ground. It is indeed known that also DAS sensitivity is strongly affected by the cable characteristic and by the coupling with the environment.

[Redacted on Author's request] we have added a sentence indicating that the lack of sensitivity may be due to the type of cable the fiber is placed in and how it is coupled to the ground, and also added a relevant citation.

Concerning the use of the full Jones matrix (in the Discussion session), in ref. [A] below it was shown that the use of the SOP vector gives the same result (in the regime where the

SOP perturbation is small enough to be useful) of the use of two of the three components of the rotation vector fully characterizing the Jones matrix in Stokes space. Indeed, the rotated Stokes parameters used in this paper, namely the deviations of S_1 and S_2 from the stationary point when the Poincare sphere is rotated such that the stationary point is the North pole, are equivalent for small deviations, to the fluctuations of the component of the rotation vector in Stokes space orthogonal to the input Stokes vector, when a reference frame rotating with the static birefringence is used (see again ref. [A] below).

We appreciate this comment and your succinct description of ref. A. We have added a sentence to the discussion describing this method and clarified how the ultimate analysis methodology is application-dependent. We have also clarified that in our analysis, the deviations of S_1 and S_2 from the North pole on the Poincare sphere are equivalent to small deviations.

Antonio Mecozzi

[A] Antonio Mecozzi, Cristian Antonelli, Mikael Mazur, Nicolas Fontaine, Haoshuo Chen, Lauren Dallachiesa, and Roland Ryf, "Use of Optical Coherent Detection for Environmental Sensing," *J. Lightwave Technol.* 41, 3350-3357 (2023).

[B] Costa, L., Varughese, S., Mertz, P. et al. "Localization of seismic waves with submarine fiber optics using polarization-only measurements," *Commun Eng* 2, 86 (2023). <https://doi.org/10.1038/s44172-023-00138-4>.

Reviewer #2

Integrated SOP sensing has been demonstrated in several contexts in recent years, and has quickly garnered attention, making this work timely. The technique used is not novel, but the main distinction of this work is the scale of its measurement campaign. The authors also propose some interesting applications regarding the improvement of network reliability, which are not commonly mentioned in the literature.

I believe this work is an interesting case-study of integrated SOP as a cost-effective and readily implementable fiber optic sensing technique and may be of interest to a broad audience, and am willing to recommend it pending some alterations to the text.

As a general suggestion, I think that in some instances the authors were slightly liberal in stating the impact of their work. For example: "In the following sections, we will demonstrate two high-impact applications that SOP sensing is primed to support: strengthening network robustness and wide-scale seismic monitoring.", "...tremendous power of SOP sensing", "...Given this high accuracy, we next evaluated the transformative potential of SOP sensing on two high-impact applications."

The impact of the method and its applications should be determined by its adoption after the fact. I think the authors could also reduce the amount of hyperbolic language.

We have removed these specific instances of hyperbolic language and toned down similar statements throughout the manuscript.

Specific comments:

Introduction:

In the first paragraph of the introduction, ref. 4 is cited twice.

We have removed the second ref. 4 in the first paragraph.

"In terms of its technical overhead, because of the larger data volume associated with DAS's high sampling-rate, DAS demands powerful computational, storage, and processing capabilities that are generally only available in high-cost systems [43, 44]." The "larger data volume" here is being presented as a disadvantage for DAS. In my view, this is an odd position: DAS has a maximum sampling rate (limited by the fiber length), but not a minimum sampling rate. In fact, it can be argued that the data-per-channel-per-acquisition of DAS is lower, since in this case for a single channel (the full integrated span of fiber), there are 3 Stokes vectors to be captured, while DAS captures directly a temperature/strain/phase/frequency shift value per sample. The high sampling rates and spatial resolution are the cause of the increased demands for storage, but all of these can be tuned to the same level as the SOP technique described. DAS has several advantages over SOP sensing but is a high-cost technique, typically demands high power launched into the fiber, is incompatible with in-line repeaters with isolators, and demands specialized narrow-linewidth lasers - these factors sufficiently justify the consideration of SOP sensing.

We agree that it's difficult to directly compare the data volume of DAS with SOP as this is highly dependent on application and sampling rates. We have therefore removed the claims of data volume throughout the manuscript.

"SOP sensing has emerged as a viable DAS alternative. Unlike DAS and interferometric systems, SOP sensing analyzes the integrated polarization changes of the modulated light traversing through traffic-carrying optical fibers [14, 15, 16]." Some recent (and not-so-recent) works have used SOP measurements without doing the full integrated fiber span. It may be more accurate to describe SOP sensing's capabilities in both localized and integrated applications, as demonstrated with P-OTDR [1], the recent publication by Fatih et al. (already referenced by the authors), and other methods like [2] and [3].

We have changed the phrasing in the introduction to mention that there are both integrated and localized uses of SOP sensing and have cited these referenced papers. We have also made a slight adjustment to the end of 3.2 (previously 2.2) to mention these works as well.

When listing contemporary sensing methods, there is no mention of inelastic scattering based techniques - I don't think the list here needs to be exhaustive, but it seems like an oversight seeing as techniques like BOTDA and ROTDR are among the most common fiber-optic sensing methods, and likely more commonplace than SOP measurements.

We have added a couple of sentences to the introduction mentioning inelastic scattering-based sensing methods (Brillouin optical time-domain analysis and Raman optical time-domain reflectometry) that are commonly used for temperature and strain measurements.

Results and Discussion:

Figure 2, I believe, should be referenced in section 2.1. It is only referenced much later in the text.

We have added a new reference to Fig. 2 in section 3.1 (formally 2.1).

Also, despite the obvious effects on the optical fibers, there seems to be no comment on ability to discriminate between different events. I don't think this is necessarily possible, unless the type of perturbations experienced is particularly constrained to predetermined type (for example, bending in [1]), but since the nonlinearity to strain is a known challenge of SOP methods over other strain-measuring techniques such as DAS, microwave, or Brillouin, it should be clearly mentioned in the text.

We have added a paragraph discussing the difficulty in discriminating between different event types by SOP alone, and present initial cross-correlation results to quantify this difficulty. Furthermore, we note that the stochastic nature of the fiber's birefringence and nonlinear response to strain make SOP sensing comparably more difficult to characterize specific stressor types than alternative methods (DAS, microwave, Brillouin).

It took me a while to understand that the correlations in figure 4 are correlations of the triggers/events, and not the acquired time-series themselves. Again, SOP signals are not linear with the perturbation, and I believe this should be more evident, when talking about the STA/LTA algorithm in section 2.2. Perhaps the authors should make this more clear in the figure caption, to prevent misunderstanding the data.

We have rephrased the caption of Fig. 4 to clarify that the correlations are between the STA/LTA events, not the raw acquired time-series data. We have also added a comment about why we leverage STA/LTA to account for the nonlinear SOP response to perturbations and mentioned this in the caption of Fig. 4.

Regarding flap detection: I thought this is an interesting application that I have not seen previously in the literature, and indeed may be a good fit since it might not require linearity and spatial resolution (such as seismology for example), and benefits from the ease-of-access of integrated SOP methods.

We appreciate this observation and added a sentence emphasizing the benefits of SOP sensing in this scenario.

Nonetheless, I think that one of the weaknesses of the presented data is that only one flap event occurred (figure 6), which makes it hard to generate a statistically significant model determining the correct thresholds for predicting the flap. It is true that this event was accompanied by an unprecedented perturbation duration, but correlation does not always imply causation, and with only one event it is hard to make any statistical claims about classification or forecasting ability. Can the authors comment on this? Is there any possible analysis of false-positive/false-negative rate that can be made?

We agree that only 1 detected flap makes it difficult to extrapolate to all future situations. We have internally discussed computing the false-positive/false-negative rate but arrived at the same conclusion that it's difficult to make any concrete statements with statistical validity. That being said, we have added a sentence to emphasize that further data collection is required to make clear statistical claims on classification or forecasting ability for diverse stressor events, and this is a high-impact opportunity to explore.

The data in figure 7 seems to support the fact that SOP can even surpass DAS in its ability to detect flaps, going by distribution of recorded events. Is there any reason for this?

We appreciate this observation and have discussed it internally. Observing the daily distribution of network flaps and SOP events, we do indeed see a greater similarity between early days of the week (e.g., 0 to 3) than with DAS events. We believe this may be due to SOP's nonlinear response to strain that contributes additional detected SOP events than DAS and warrants further exploration in future work. We have added a note mentioning this.

In Figure 9, it seems that the authors only report earthquakes very close to the fibers, and of low magnitudes. Given the length of the measurement campaign, has there been any demonstration of measurement of larger earthquakes, at farther distances? That might be a more interesting scenario as a case-study for early warning, as this application is predicated on the ability to sense P-waves.

We appreciate this comment suggesting we expand the seismic sensing radius beyond the reported 100km. When we initially performed the analysis, we considered a larger range as well but saw no significant perturbations in this expanded range. Consequently, we decided to only report events within a smaller 100km radius which is comparable to the sensing range of other EEW systems and beyond the 40-60km blind zone of typical EEW systems. That being said, we have added additional references and a paragraph clarifying how these insights may still be used for future seismic monitoring applications.

Related to this, I think the following statement in the discussion is not necessarily true: "Third, we explored the possibility of leveraging existing fiber-optic networks for terrestrial seismic sensing, demonstrating the potential to increase Southern California's seismic monitoring coverage by 150%. Combined with a thorough analysis elucidating the impact of seismic waves on terrestrial fiber, we have demonstrated how optical networks can aid in environmental monitoring and earthquake early warning systems." Indeed, optical networks can benefit environmental monitoring, but for earthquake early warning, using SOP, I don't think the data presented in this paper supports that case.

We have removed this claim, and in general, toned down the summary statement to be more factual based on the presented evidence.

Discussion:

"Enabling such spatial localization with SOP sensing would increase its practicality and transform it into a full-fledged DAS replacement.". Some works (see refs [1],[2],[3] and Fatih et al. work) have shown spatial localization using SOP sensing (although not in terrestrial networks).

We have changed the phrasing in the introduction to mention that there are both integrated and localized uses of SOP sensing and have cited the referenced papers. We have also made a slight adjustment to the end of 3.2 to mention these works as well.

More importantly, SOP is not set-up to be a DAS replacement, even with localization: DAS has a linear response to applied strain, and seems to be much more sensitive, even with comparably short gauge lengths.

We have removed the claim that SOP is a DAS replacement, and instead phrased it as an alternative depending on the context.

[1] - Distributed polarimetric measurements for optical fiber sensing, Luca Palmieri, Optical Fiber Technology, 19, 6, 2013, 720-728

[2] - Costa, L., Varughese, S., Mertz, P. et al. Localization of seismic waves with submarine fiber optics using polarization-only measurements. Commun Eng 2, 86 (2023).

[3] - A. Galtarossa, D. Grosso, L. Palmieri and L. Schenato, "Reflectometric Characterization of Hinges in Optical Fiber Links," in IEEE Photonics Technology Letters, vol. 20, no. 10, pp. 854-856, May15, 2008

Reviewer #3

Review Letter

Dear Authors and Editor, thank you for having submitted the your valuable work to Nature Communications Engineering. The paper is focused on the exploitation of the terrestrial telecommunications networks infrastructure for environmental sensing leveraging on state of polarization (SOP) monitoring obtained by deployed transceivers. This idea is gaining interest not only in the telecommunications/optical community, but also in the earth sciences community as its implementation would, at the same time, double the optical networks value offering additional services and open the doors for immense dataset to the geophysics people. Moreover, while several papers have been published on this topic, limited to subsea optical links, less work has been done on terrestrial networks. In the reviewer's opinion this already represents an important selling point for this work. Indeed, terrestrial sensing offers several advantages (and disadvantages) w.r.t. subsea sensing. Subsea communications are commonly point-to-point-links and SOP can be sampled only at transmitter/receiver sites and it's very difficult to add further devices and SOP monitoring devices as they are placed underwater.

Terrestrial networks are instead usually arranged in several topologies as ring or mesh, and it is easier to access and upgrade transceivers, add/drop sites or add additional sensors. The possibility to have more sensing points across the network can counterbalance the fact that SOP sensing provides integrated measurements in space w.r.t. to DAS and cover wider areas. This is clearly stated in the paper, and I

really appreciate the authors to clearly show to the reader a rough estimation of how much the traditional sensor network considered can be enlarged integrating SOP sensing. On the other side, terrestrial SOP sensing is afflicted by a much larger amount of noise w.r.t. the underwater scenario, generated by several anthropic activities, especially in a largely crowded area.

In this context, a paper like this, based on a solid long term observation (86 days) of real deployed devices compared to equivalent parallel fiber DAS observation, surely deserves publication on this journal. However, apart from some minor aspects that can be adjusted or clarified, I have some concerns about the generality of the main findings, although I understand that such experimental work on a such novel subject may not always scale to find general rules. I will discuss the main points in the following:

I would move section 4 straight after the Introduction section. This is substantially a section where the authors explain the data retrieval and post-processing methodology. This would make much clearer to the reader the following discussion. **We have moved the Methods section after the Introduction to explain the data retrieval and post-processing methodology as early as possible.**

Also, supplementary figures can be included in the main paper flow for better readability.

We have moved the supplementary figures into the main paper to improve the flow.

Also, if not covered by NDA, the author may report the used transceiver model and the characteristics in terms of sampling rate and the accuracy of the Stokes vector measurement (if available) provided.

We agree with this comment, however, Google has explicitly asked us not to release this information in the publication.

In section 2 the authors discuss how different fiber stressors (bending, moving, etc.) impact Stokes vector and they report one evolution per type of event. It is known that polarization evolution depends on the stochastic realization of the fiber birefringence. My biggest concern here is about the repeatability and distinguishability of those typical waveform. In my opinion the authors should clarify to what extent the obtained waveform for a specific stressor can be more or less strictly identified by some features of typical evolution and how much those waveforms are distinguishable among the different stressors, for example, by means of crosscorrelation analysis.

We have added a paragraph discussing the difficulty in discriminating between different event types by SOP alone, and present cross-correlation results to quantify this difficulty. We note that the stochastic nature of the fiber's birefringence and nonlinear response to strain make SOP sensing comparably more difficult to characterize specific stressor types compared to alternative methods (DAS, microwave, Brillouin).

In section 2.2 is not clear what Q1, Q3 values mean. The whole explanation about the crosscorrelation lags appears in general not clear.

We have added the definition for Q1 and Q3 and attempted to clear up the cross-correlation lag discussion.

Also, the reference base of time used should be clarified. From Supplementary Fig.1 caption it seems that the SOP perturbation is detected prior to DAS (-5/-6 seconds) for the same event and that this lag increases due to timestamp drifts.

We have added a comment indicating which signal is used as the reference signal.

Need for precise time synchronization in data network sensing is a known issue.

We have added a note that precise time synchronization is a common issue in data network sensing

At the end of section 2.2, its not necessarily true that SOP sensing has necessarily low data storage requirements. Applications outside the real time early warning service, such as recording SOP waveform for offline geophysical studies, may require a lot storage depending on also on the SOP sampling rate.

We agree that it's difficult to directly compare the data volume of DAS with SOP as this is highly dependent on application and sampling rates. We have therefore removed the claims of data volume throughout the manuscript.

There has been some investigation on the localization using SOP sensing by the group of E.Virgillito, R. Bratovich et al. [1], even when coherent transceivers are not available. This can be done by performing SOP sensing on opposite direction. You may cite [1] to mention this.

We have changed the phrasing in the introduction to mention that there are both integrated and localized uses of SOP sensing and have cited the referenced paper. We have also made a slight adjustment to the end of 3.2 to mention this as well.

In section 2.3 the authors take over on the usage of SOP sensing for network reliability. In this case I would doubt the generality of the approach given that you observed 17 non-flap events with magnitude larger than 6.9. As the authors mention, the interplay between the Stokes can be significant. To this aim, I would suggest considering an aggregate metric such as the SOP change rate, which sums the variations of S1, S2, S3, other than more complex classifiers such as ML-based as suggested by the authors.

We appreciate your suggestion on using the SOP change rate over more complex classifiers and added a paragraph at the end of 3.3.1 discussing this as a future direction.

Although the reporting of the single flap event is quite interesting, it seems too limited to derive thresholds for preemptive rerouting. From section 2.3.2 indeed is not clear if in the temporal distribution also non-flap events are included. If only events corresponding to an effective network flap are reported, comparing the SOP waveform characteristic to the single event reported in section 2.3.1 would be interesting.

[Redacted on Author's request] Since the SOP/DAS distributions contain non-flap events, we agree with the reviewer that it would be difficult to compare the SOP waveform characteristics to the previously detected flap. However, we have explicitly called out the fact that the detected flap occurred during a time of day corresponding to high-flap activity across the network to make our point clearer.

Also, in section 2.3.2, it would be useful to mention the source for the global flaps data.

We have added a slight description of the internal tool used to monitor global flap data, [Redacted on Author's Request]

Section 2.4 is then demanded to the analysis of the collected data for seismic monitoring. Regarding the scalability of the SOP monitoring approach on coherent transceivers, practical issues on accessing Stokes vector should be mentioned: while the authors had the opportunity to access the internal registers of the coherent transceivers, most of these transceivers from different vendors do not expose this data due to the closed source nature of the DSP, although surely available at low level, and it is not straightforward for whatever network operator to obtain access to them after deploying in production network due to constraints imposed by transceiver vendors and technical limitations. In [1] the authors proposed an approach to overcome these limitations: SOP data can be monitored from intensity modulated – direct detected (IMDD) data signals using cheaper SOP monitoring devices based on polarization beams splitters. Although this requires additional devices and expenditures, this approach can leverage on a wide number of IMDD sources, such as optical supervisory channels present at many amplification sites or transceiver in the access network segments which still largely operated with IMDD transceivers.

This is an excellent observation, and we have added a new paragraph to the discussion section that mentions (a) our unique access to the internal DSP register and (b) the feasibility of using additional equipment at various points in the network for SOP sensing in the absence of direct access. Additionally, we have cited the proposed paper.

I suppose that origin time E in Fig.9/10 is given by the seismograms station. Please clarify this.

We have added a comment to Figs. 9 and 10 mentioning that E is given by the seismogram stations.

The authors state that the aerial section of the fiber act as an amplifier for the incoming P/S waves and support this statement with discussion based on Supplementary Fig.2. I agree that strong fiber shaking due to wind gusts would trigger large SOP oscillations. However, I am not really convinced that such oscillations can amplify SOP oscillations due to seismic waves, especially since fiber aerial sections are not directly coupled to the ground motion. Moreover, independently on the sampling rate of the internal DSP polarimeter offered to the final user, the actual sampling rate used for data signal reception is usually of the same order of magnitude of the optical channel symbol rate, which should be between 32 and 96 GBaud (10^9 Hz order) in state of the art transceivers, to successfully track data signal. Hence, tolerance to SOP oscillation rate should be typically far slower than the sampling rate, otherwise the receiver would lose the lock on the received channel leading to a potential network flap. Hence, this would suggest that strong SOP oscillations observed may be linked to fiber oscillations caused by wind, which show their effect on the integrated fiber length and that the amplification mechanism needs further evidence from the authors.

We appreciate the reviewer's comments on our hypothesis surrounding the aerial fiber section acting as an amplifier for seismic-induced perturbations, as well as the clearer explanation of the DSP tracking rate in regard to the optical channel symbol rate and tolerance to oscillations. We have addressed this comment in two ways. First, for the sake of clarity, we have removed the discussion on the DSP's tracking rate to instead focus on the potential impact of wind. Second, we added a citation that outlines how different structures — that may mount aerial fiber — respond to seismic activity. Third, to better rule out the impact of wind during the 3 periods of seismic activity, we mention that there were significantly lower wind speeds and wind gusts during all collection periods.

This point leads to the more general topic of how SOP oscillations caused by seismic waves can be distinguished from other sources, such as wind gusts for aerial sections or possible fiber maintenance activity, which is my main concern about the presented claims. I am quite surprised to not see a larger amount of noise caused by other anthropic activities in the P wave segments considering that the optical link under observation is deployed in a highly populated area, which may also partially explain why among the 478 candidate earthquakes only less than 1% have

shown meaningful SOP perturbations. Possibly, machine learning based approaches as in [2] may improve the situation. In this context, I agree that surface wave sensing is still an attractive opportunity if used in a smart grid context where many optical links, farther from highly populated areas, can be used simultaneously to detect incoming earthquake and save precious seconds for early warning.

We have added a paragraph that acknowledges the difficulty of reliably distinguishing between SOP waveforms, and cite the referenced paper as an example of more advanced machine learning methods being used to improve classification accuracy. We further clarify that using these more advanced methods, surface wave sensing could still be an attractive opportunity when used in a "smart grid context."

Best Regards,
The Reviewer

[1] E. Virgillito et al., "Detection and Localization of Metropolitan Anthropogenic Activities by SOP Monitoring of IM-DD Optical Data Channels," in 2023 International Conference on Photonics in Switching and Computing (PSC), Sep. 2023, pp. 1–3. doi:

10.1109/PSC57974.2023.10297183.

[2] Z. Li, M.-A. Meier, E. Hauksson, Z. Zhan, and J. Andrews, "Machine Learning Seismic Wave Discrimination: Application to Earthquake Early Warning," Geophysical Research Letters, vol. 45, no. 10, pp. 4773–4779, 2018, doi: 10.1029/2018GL077870.

Decision letter and referee reports: second round

29 Apr 2024

Dear Mr Carver,

Your manuscript titled "Polarization sensing of network health and seismic activity over a live terrestrial fiber-optic cable" has now been seen by our referees, whose comments appear below. In light of their advice I am delighted to say that we are happy, in principle, to publish a suitably revised version in Communications Engineering under the open access CC BY licence (Creative Commons Attribution v4.0 International Licence).

We therefore invite you to revise your paper one last time to address the remaining concerns of our reviewers. At the same time we ask that you edit your manuscript to comply with our journal policies and formatting style in order to maximise the accessibility and therefore the impact of your work.

EDITORIAL REQUESTS

- * Your manuscript should comply with our policies and format requirements, detailed in our style and formatting guide (<https://www.nature.com/documents/commsj-phys-style-formatting-guide-accept.pdf>).
- * Please edit your manuscript according to the editorial requests in the attached table, and outline revisions made in the right hand column. If you have any questions or concerns about any of our requests, please do not hesitate to contact me. It is important that each request be addressed in order to avoid delays in accepting your manuscript. Please upload the completed table with your manuscript files.
- * Please refer to our checklist for a full list of the files that must be provided upon resubmission: <https://www.nature.com/documents/commsj-file-checklist.pdf> Please ensure a completed version of this file is uploaded as a Related Manuscript with your final submission. (The file checklist is also provided at the end of the editorial requests table)
- * An updated editorial policy checklist that verifies compliance with all required editorial policies must be completed and uploaded with the revised manuscript. All points on the policy checklist must be addressed; if needed, please revise your manuscript in response to these points. Please note that this form is a dynamic 'smart pdf' and must therefore be downloaded and completed in Adobe Reader. Clicking this link will download a zip file containing the pdf.

OPEN ACCESS

Communications Engineering is a fully open access journal. Articles are made freely accessible on publication under a CC BY license (Creative Commons Attribution 4.0 International License). This license allows maximum dissemination and re-use of open access materials and is preferred by many research funding bodies.

For further information about article processing charges, open access funding, and advice and support from Nature Research, please visit <https://www.nature.com/commseng/about/open-access>

RESUBMISSION

Please refer to the checklist at the end of the editorial request table for a full list of the files that must be provided upon resubmission.

Decision letter and referee reports: second round

At acceptance, the corresponding author will be required to complete an Open Access Licence to Publish on behalf of all authors, declare that all required third party permissions have been obtained and provide billing information in order to pay the article-processing charge (APC) via credit card or invoice.

Please use the following link to submit your revised files:

[REDACTED]

We hope to hear from you within two weeks; please let us know if the process may take longer.

Sincerely,

Anastasiia Vasylchenkova
Communications Engineering
Associate Editor

REVIEWERS' COMMENTS:

Reviewer #1 (Remarks to the Author):

The authors satisfactorily responded to my comments. I suggest publication.

Reviewer #2 (Remarks to the Author):

The authors have adequately addressed all of my comments (and, in my opinion, the comments of the other reviewers as well). I support the publication of the paper in its current form.

Reviewer #3 (Remarks to the Author):

Dear Authors,
thank you for having accepted the reviewer comments and improved the quality of your manuscript. I recommend the article to be accepted for publication. I have just a small note. In the newly added paragraph at page 14, I would suggest to remove "inexpensive" prior to polarimeters. While polarization beam splitters are a cheap option, polarimeters are definitely more expensive.

To all reviewers, we sincerely thank you for taking the time to read our original manuscript, offer suggestions for improvement, and confirm our final edits.

Reviewer #1

The authors satisfactorily responded to my comments. I suggest publication.

Reviewer #2

The authors have adequately addressed all of my comments (and, in my opinion, the comments of the other reviewers as well). I support the publication of the paper in its current form.

Reviewer #3

thank you for having accepted the reviewer commentsa and improved the quality of your manuscript. I recommend the article to be accepted for publication. I have just a small note. In the newly added paragraph at page 14, I would suggest to remove "inexpensive" prior to polarimeters. While polarization beam splitters are a cheap option, polarimeters are definitely more expensive.

This is a fair statement, and we have removed the word inexpensive in this context.